



# A WRF-Chem study of the greenhouse gas column and in situ surface mole fractions observed at Xianghe, China. Part 2: Sensitivity of carbon dioxide (CO$_2$) simulations to critical model parameters

Sieglinde Callewaert[1,4], Minqiang Zhou[1,2], Bavo Langerock[1], Pucai Wang[3], Ting Wang[3], Emmanuel Mahieu[4], and Martine De Mazière[1]

[1]Royal Belgian Institute for Spacy Aeronomy (BIRA-IASB), Brussels, Belgium
[2]State Key Laboratory of Atmospheric Environment and Extreme Meteorology, Institute of Atmospheric Physics, Chinese Academy of Sciences, Beijing, China
[3]CNRC & LAGEO, Institute of Atmospheric Physics, Chinese Academy of Sciences, Beijing, China
[4]UR SPHERES, Department of Astrophysics, Geophysics and Oceanography, University of Liège, Liège, Belgium

**Correspondence:** Sieglinde Callewaert (sieglinde.callewaert@aeronomie.be), Minqiang Zhou (minqiang.zhou@aeronomie.be)

**Abstract.** Understanding the variability and sources of atmospheric CO$_2$ is essential for improving greenhouse gas monitoring and model performance. This study investigates temporal CO$_2$ variability at the Xianghe site in China, which hosts both remote sensed (TCCON-affiliated) and in situ (PICARRO) observations. Using the Weather Research and Forecast model coupled with Chemistry, in its greenhouse gas option (WRF-GHG), we performed a one-year simulation of surface and column-averaged CO$_2$ mole fractions, evaluated model performance and conducted sensitivity experiments to assess the influence of key model configuration choices. The model captured the temporal variability of column-averaged mole fraction of CO$_2$ (XCO$_2$) reasonably well (r=0.7), although a persistent bias in background values was found. A July 2019 heatwave case study further demonstrated the model's ability to reproduce a synoptically driven anomaly. Near the surface, performance was good during afternoon hours (r=0.75, MBE=–1.65 ppm), nighttime mole fractions were overestimated (MBE = 6.51 ppm), resulting in an exaggerated diurnal amplitude. Sensitivity tests revealed that land cover data, vertical emission profiles, and adjusted VPRM-parameters(Vegetation Photosynthesis and Respiration Model) can significantly influence modeled mole fractions, particularly at night. Tracer analysis identified industry and energy as dominant sources, while biospheric fluxes introduced seasonal variability - acting as a moderate sink in summer for XCO$_2$ and a net source in most months near the surface. These findings demonstrate the utility of WRF-GHG for interpreting temporal patterns and sectoral contributions to CO$_2$ variability at Xianghe, while emphasizing the importance of careful model configuration to ensure reliable simulations.

## 1 Introduction

Climate change is one of the most pressing global challenges, and carbon dioxide (CO$_2$) is its primary driver due to its long atmospheric lifetime and rising atmospheric abundance (Masson-Delmotte et al., 2021). Understanding how atmospheric



$CO_2$ levels vary over time and space is essential for detecting long-term trends, distinguishing natural fluctuations from an-
thropogenic signals, and deepening our insight into the carbon cycle and its interactions with the atmosphere. Observational
records are key to unraveling local and regional carbon budgets and assessing the effectiveness of mitigation strategies. To fully
interpret such observations, especially in complex environments, we rely on atmospheric transport models, which provide spa-
tial and temporal context and help disentangle the observed $CO_2$ signal into contributions from different sources and processes.
As the world's largest fossil $CO_2$ emitter (Friedlingstein et al., 2025), our study focuses on China—a country whose vast and
densely populated regions, strong industrial activity, and ecological diversity make it a complex but highly relevant area for
atmospheric $CO_2$ research. In this context, a ground-based remote sensing instrument was installed in 2018 at the Xianghe site,
a suburban location approximately 50 km southeast of Beijing. The Fourier Transform Infrared (FTIR) spectrometer provides
high-precision column-averaged $CO_2$ mole fractions and is part of the global Total Carbon Column Observing Network (TC-
CON). Complementing this, a PICARRO Cavity Ring-Down Spectroscopy (CRDS) analyzer measures near-surface $CO_2$ mole
fractions at 60 meters above ground level. This unique combination of collocated column and in situ observations— to our
knowledge currently the only such setup in China—offers a valuable opportunity to study both local and regional $CO_2$ signals
and to evaluate model performance for different levels of the atmosphere.

Previous studies by Yang et al. (2020, 2021) provided initial insights into the seasonal and diurnal variability of both column-
averaged and near-surface $CO_2$ mole fractions at Xianghe. Their work highlighted the strong influence of local and regional
emissions, as well as planetary boundary layer dynamics, on observed $CO_2$ levels. However, these analyses were either purely
observational or relied on coarser-resolution model products such as CarbonTracker, which are limited in their ability to re-
solve mesoscale variability. Furthermore, the two observation types were not jointly analyzed within a high-resolution modeling
framework, leaving room for a more detailed and integrated approach. To gain a deeper understanding of the processes shaping
the observed $CO_2$ mole fractions at Xianghe, we apply the high-resolution WRF-GHG model in this work, a specific config-
uration of the widely used WRF-Chem model tailored for greenhouse gas simulations. The current study is part of a broader
research effort investigating multiple greenhouse gases at the site. While a companion paper has already presented the results
for $CH_4$ (Callewaert et al., 2024), the present work focuses exclusively on $CO_2$.

The WRF-GHG model was originally developed to address the limitations of coarse-resolution global models by providing a
more detailed representation of $CO_2$ transport, surface flux exchanges, and meteorological processes at the mesoscale. Thanks
to its coupling with the Vegetation Photosynthesis and Respiration Model (VPRM), WRF-GHG has demonstrated strong ca-
pabilities in simulating biogenic $CO_2$ fluxes and atmospheric dynamics. It has been successfully applied across a range of
environments, from rural areas influenced by sea-breeze circulations (Ahmadov et al., 2007, 2009) to urban regions with com-
plex emission patterns and boundary layer processes (Feng et al., 2016; Park et al., 2018; Zhao et al., 2019). Further, the model
has been evaluated against in situ, tower, aircraft and satellite data during large-scale campaigns such as ACT-America in the
US (Hu et al., 2020) and KORUS-AQ in South-Korea (Park et al., 2020), showing its ability to reasonably capture spatiotem-
poral variability of $CO_2$. In China, WRF-GHG has been used to study $CO_2$ fluxes and atmospheric mole fractions on a national
scale and to explore the role of biospheric and anthropogenic sources (Dong et al., 2021; Ballav et al., 2020). Li et al. (2020)
evaluated WRF-GHG against tower observations in northeast China, showing the model could capture seasonal trends and



episodic enhancements, despite underestimating diurnal variability and respiration fluxes.

Our study uses WRF-GHG to investigate the main drivers of observed temporal variations at Xianghe and to evaluate the model's ability to reproduce these patterns, identifying key sources of error where relevant. The model's tracer framework further allows us to disentangle the contributions of anthropogenic, biogenic, and meteorological processes to simulated $CO_2$ levels. The structure of the paper is as follows: Section 2 describes the observations, model configuration, and the design of additional model sensitivity experiments. Section 3 presents the results, including model performance, tracer-based analyses and sensitivity experiments. Section 4 summarizes the conclusions and outlook.

## 2 Methods

### 2.1 Observations at Xianghe site

We use observational data from the atmospheric monitoring station situated in Xianghe county (39.7536° N, 116.96155° E; 30 m a.s.l.). This site is located in a suburban part of the Beijing-Tianjin-Hebei (BTH) region in northern China. The town center of Xianghe lies approximately 2 km to the east, while the major metropolitan areas of Beijing and Tianjin are situated roughly 50 km to the northwest and 70 km to the south-southeast, respectively (Fig. 1b). The dominant vegetation in the surrounding area consists of cropland.

Continuous atmospheric measurements have been conducted at the Xianghe observatory by the Institute of Atmospheric Physics (IAP), Chinese Academy of Sciences (CAS), since 1974. FTIR solar absorption measurements have been performed since June 2018, from the roof of the observatory by a Bruker IFS 125HR. This ground-based remote sensing instrument records spectra in the infrared range and is part of the TCCON network (Wunch et al., 2011; Zhou et al., 2022), providing data on total column-averaged dry air mole fractions for gases such as $CO_2$, $CH_4$, and CO (noted as $XCO_2$, $XCH_4$ and XCO, respectively). The current study employs the GGG2020 data product (Laughner et al., 2024). Observations are typically taken every 5 to 20 min, depending on weather conditions and instrument status. TCCON measurements are exclusively performed under clear skies. The uncertainty associated with the $XCO_2$ measurements is approximately 0.5 ppm. Further details regarding the instrument and the retrieval methodology can be found in Yang et al. (2020).

In addition to the FTIR measurements, in situ measurements of $CO_2$ and $CH_4$ mole fractions have been conducted since June 2018 using a PICARRO cavity ring-down spectroscopy G2301 analyzer. This instrument draws air from an inlet situated on a 60 m tower. A more comprehensive description of this measurement setup is available in Yang et al. (2021). The measurement uncertainty for $CO_2$ with this instrument is about 0.06 ppm. The data used in this study were converted to align with the WMO $CO_2$ X2019 scale (Hall et al., 2021).

### 2.2 WRF-GHG model simulations

We make use of the WRF-GHG model simulations elaborated in Part 1 of this work (Callewaert et al., 2024), and provide a brief summary here for completeness. The simulations were performed using the Weather Research and Forecasting model





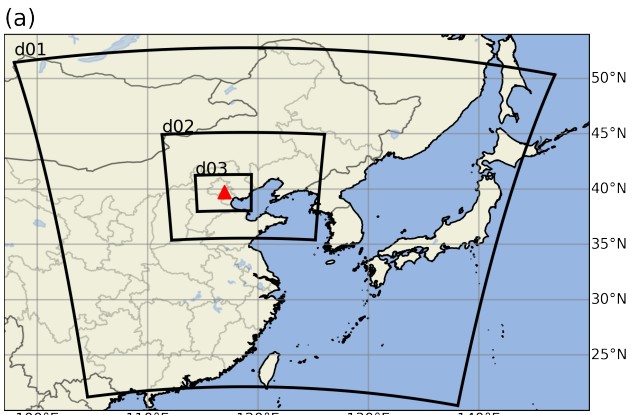
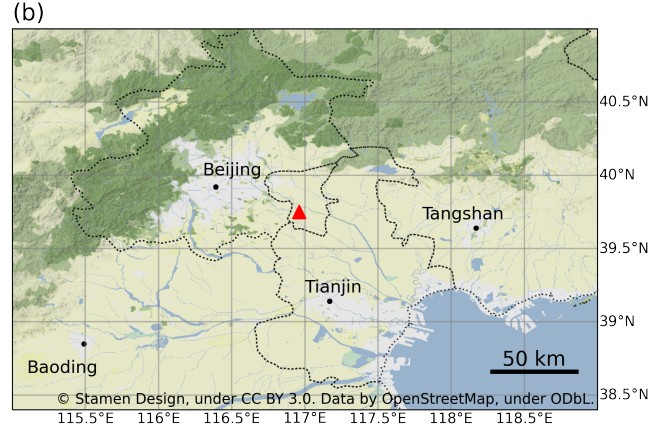

**Figure 1.** (a) Location of the WRF-GHG domains, with horizontal resolutions of 27 km (d01), 9 km (d02) and 3 km (d03). All domains have 60 (hybrid) vertical levels extending from the surface up to 50 hPa. (b) Terrain map including the largest cities in the region of Xianghe, roughly corresponding to d03. The location of the Xianghe site is indicated by the red triangle in both maps. Figure taken from Callewaert et al. (2024).

with Chemistry (WRF-Chem v4.1.5; Grell et al. (2005); Skamarock et al. (2019); Fast et al. (2006)) in its greenhouse gas configuration, called WRF-GHG (Beck et al., 2011). This Eulerian transport model simulates three-dimensional greenhouse gas mole fractions simultaneously with meteorological fields, without accounting for chemical reactions. The model setup includes three nested domains with horizontal resolutions of 27 km, 9 km, and 3 km, and 60 vertical levels extending from the surface up to 50 hPa (Figure 1a).

Anthropogenic $CO_2$ emissions were obtained from CAMS-GLOB-ANT v5.3 (Granier et al., 2019; Soulie et al., 2023) and temporally disaggregated using CAMS-TEMPO profiles (Guevara et al., 2021). The original 11 source sectors were aggregated into four broader categories and included in separate tracers: energy, industry, transportation, and residential & waste. Biomass burning emissions were taken from the FINN v2.5 inventory (Wiedinmyer et al., 2011), and ocean-atmosphere $CO_2$ fluxes were prescribed based on the climatology from Landschützer et al. (2017). Finally, biogenic $CO_2$ fluxes were calculated online using VPRM (Mahadevan et al., 2008; Ahmadov et al., 2007), driven by WRF-GHG meteorology and MODIS surface reflectance data, with ecosystem-specific VPRM parameters from Li et al. (2020) and land cover information from SYNMAP (Jung et al., 2006).

Meteorological fields (e.g., wind, temperature, humidity) were driven by hourly data from the European Centre for Medium-Range Weather Forecasts (ECMWF) ERA5 hourly data ($0.25° \times 0.25°$; Hersbach et al. (2023a, b)). Daily restarts were performed at 00:00 UTC, with model initialization at 18:00 UTC the day before to allow for a 6-hour spin-up, stabilizing the simulation. For tracer fields, mole fractions at 00:00 UTC were copied from the previous day's simulation to maintain consistency. The initial and lateral boundary conditions for $CO_2$ were prescribed using the 3-hourly Copernicus Atmosphere Monitoring Service (CAMS) global reanalysis (EGG4, Agustí-Panareda et al. (2023)).

The final simulated $CO_2$ field is composed of the sum of several tracers that track contributions from individual sources. These





include a background tracer (reflecting the evolution of initial and lateral boundary conditions from CAMS), as well as tracers for energy, industry, residential, transportation, ocean, biomass burning, and biogenic fluxes.

WRF-GHG was run from 15 August 2018 to 1 September 2019. However, the first two weeks were regarded as a spin-up phase, so the analysis in Sect. 3.1 is made on one full year of data: from 1 September 2018 until 1 September 2019. The complete data set can be accessed on https://doi.org/10.18758/P34WJEW2 (Callewaert, 2023).

To enable comparison with the observations, model data from the grid cell containing the measurement site are extracted. For near-surface observations, the model profile is interpolated to the altitude of the instrument, while for column measurements the model output is smoothed with the FTIR retrieval's a priori profile and averaging kernel. Further details are provided in Part 1 (Callewaert et al., 2024).

## 2.3   Sensitivity experiment design

A series of sensitivity experiments was conducted to assess the impact of key model assumptions on surface $CO_2$ fluxes, such as the treatment of emission heights, land cover classification, and biogenic flux parameterizations. Four two-week periods were selected for these sensitivity simulations, spanning from the 15th to the 29th of March, May, July, and December. These months were identified as being most critical for simulating the diurnal $CO_2$ cycle while representing different seasons. Although the specific dates within each month (15–29) were chosen somewhat arbitrarily, they were applied consistently across all four

120   months to ensure comparability. Three different simulation experiments (BASE, LC, PARAM) were performed over these four periods to isolate the impact of three model assumptions (see Table 1):

- **Emission height.** To assess the impact on simulated in situ $CO_2$ mole fractions at Xianghe of the height at which anthropogenic emissions are released in the atmosphere (all at the lowest model level near the surface, or according to sector-specific vertical profiles), we applied the vertical profiles for point sources from Brunner et al. (2019) to the

CAMS-GLOB-ANT sector-specific $CO_2$ emissions. For the fluxes in the 'industrial processes' sector, we used the average of the profiles of SNAP 3 (Combustion in manufacturing industry) and SNAP 4 (Production processes). Note that we don't make a distinction between area and point sources as in Brunner et al. (2019), as this information is not available for our study region. Both SFC (surface) and PROF (profile) emissions were included in all experiments as separate tracers, enabling direct comparison of the impact of emission height on simulated $CO_2$ at Xianghe.

- **Land cover classification.** The biogenic $CO_2$ fluxes are calculated online in WRF-GHG as the weighted average of the Net Ecosystem Exchange (NEE) for eight vegetation classes (evergreen trees, deciduous trees, mixed trees, shrubland, savanna, cropland, grassland and non-vegetated land)(Mahadevan et al., 2008). As a default, the SYNMAP land cover map is used to calculate the vegetation fraction for every model grid cell. To assess the impact of this classification, we prepare the VPRM model input files for the LC and PARAM experiments with the global 100-m Copernicus Dynamic

Land Cover Collection 3 (epoch 2019) (Buchhorn et al., 2020), using the pyVPRM python package (Glauch et al., 2025), allowing for an updated and higher-resolution representation of vegetation types in the domain.





**Table 1.** Overview of the model configuration for the sensitivity experiments. Note that the Glauch parameter table does not include values for the 'Savanna' class, consistent with its absence in the Copernicus land cover map.

| Name | Emission height | Land cover map | VPRM parameters |
|------|-----------------|----------------|-----------------|
| BASE | SFC, PROF | SYNMAP | Li table |
| LC | SFC, PROF | Copernicus LC | Li table |
| PARAM | SFC, PROF | Copernicus LC | Glauch table |

– **VPRM parameterization.** The VPRM-calculated NEE can be tuned for different regions around the globe by specifying four empirical parameters ($\alpha$, $\beta$, $\lambda$ and $PAR_0$) per vegetation class. These parameter tables are a mandatory input to the WRF-GHG model and can be calibrated using a network of eddy flux tower sites, representing the different vegetation classes in the region, or taken from literature. Due to the lack of a dedicated calibration study in China, we applied the table from Li et al. (2020) in the one-year simulations, and the BASE and LC experiments. Seo et al. (2024) reported the lowest RMSE in East Asia using these parameter values, relative to the default US settings and those of Dayalu et al. (2018). To evaluate the impact of these parameters at Xianghe, we conducted an experiment (PARAM) with an alternative parameter table, optimized over Europe by Glauch et al. (2025). The exact parameter values used in each experiment are provided in Table A1.

By comparing the results from the three sensitivity experiments, the influence of individual model components can be isolated. The impact of land cover classification is assessed by comparing the BASE and LC simulations, which differ only in the land cover dataset used. Similarly, the effect of VPRM parameterization can be evaluated by comparing LC and PARAM, which share the same land cover input but differ in the VPRM parameter table. Finally, the role of vertical emission distribution can be assessed using the additional tracers in each simulation and by comparing results across all three experiments. This modular approach enables a systematic investigation of potential model deficiencies affecting the representation of $CO_2$ at Xianghe. Note that the BASE experiment with SFC emissions corresponds exactly with the settings of the one-year simulation described above.

## 3 Results

### 3.1 Evaluation of one-year simulation

#### 3.1.1 Timeseries and statistical comparison

Figure 2 provides an overview of the simulated and observed time series of $CO_2$ mole fractions at Xianghe, along with their respective differences. Overall, WRF-GHG demonstrates a reasonable accuracy in replicating these measurements: the $XCO_2$ observations are slightly underestimated, with a mean bias error of -1.43 ppm (as detailed in Table 2) and a correlation coefficient of 0.70. Note that the $XCO_2$ time series was de-seasonalized before calculating the correlation coefficient in order





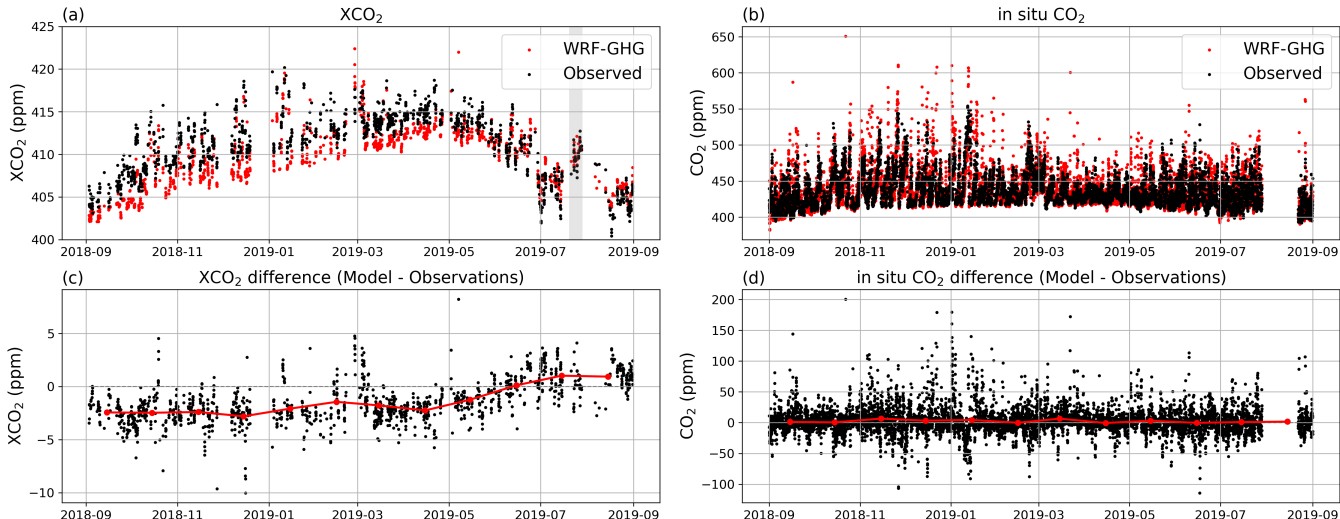

**Figure 2.** Time series of the observed (black) and simulated (red) (a) $XCO_2$ and (b) insitu $CO_2$ mole fractions at the Xianghe site. Panels (c) and (d) show the differences between WRF-GHG simulations and observations for $XCO_2$ and in situ $CO_2$, respectively. Data points are hourly. The red data points in (b) and (d) represent the monthly mean differences.

**Table 2.** Statistics of the model-data comparison of the ground-based $CO_2$ observations at the Xianghe site from 1 September 2018 until 1 September 2019. We present the mean bias error (MBE), root mean square error (RMSE) and Pearson correlation coefficient (CORR). The MBE and RMSE are given in ppm. For in situ observations, the data is split in afternoon (13-18 LT) and night (3-8 LT) hours.

|       | insitu $CO_2$ (afternoon) | insitu $CO_2$ (night) | $XCO_2$ |
|-------|---------------------------|-----------------------|---------|
| MBE   | -2.34                     | 5.82                  | -1.43   |
| RMSE  | 14.54                     | 24.88                 | 2.45    |
| CORR  | 0.75                      | 0.58                  | 0.70    |

to remove the effect of the seasonal variation. The data near the surface has been divided into afternoon (13:00 - 18:00 LT) and nighttime (03:00 - 08:00 LT) periods to evaluate model performance, as these time frames often yield different results. Indeed, WRF-GHG shows a smaller bias (-2.34 ppm) during the afternoon, when the lower atmosphere is well-mixed, compared to nighttime (5.82 ppm). Additionally, the MBE differs in sign between the two periods: near-surface $CO_2$ levels tend to be underestimated by the model in the afternoon but overestimated at night. Except for the moderate correlation observed for in situ $CO_2$ during nighttime (0.58), WRF-GHG achieves relatively high correlation coefficients ($\geq 0.7$) for other $CO_2$ data, indicating satisfactory model performance.

A few aspects stand out when looking at the time series of $XCO_2$ in Fig. 2a,c. Firstly, a clear model underestimation is observed between September 2018 and May 2019, after which the bias diminishes. Secondly, from May onward, $XCO_2$ generally declines, which is associated with the northern hemisphere's growing season and enhanced photosynthesis from the biosphere. However, a significant spike in $XCO_2$ levels occurs between 20-29 July, highlighted in gray in Fig. 2a. A more





detailed analysis of this summer spike will be provided in Sect. 3.3, while the bias between September 2018 and May 2019 is discussed below (Section 3.1.2).

### 3.1.2 Correction of background bias

WRF-GHG underestimates $XCO_2$ by approximately 2 ppm until May 2019, after which the negative bias diminishes. This bias likely originates from a similar error in the background data, inaccuracies in representing the actual sources and sinks in the region, or a combination of both.

The CAMS validation report (Ramonet et al., 2021) presents "a very good agreement for all (TCCON) sites", suggesting that the CAMS reanalysis that is driving the WRF-GHG simulations is of good quality without known biases. However, their crite-
ria for what constitutes "very good" appears to be relatively mild (within $\pm 2$ ppm). Moreover, the Xianghe site wasn't included in this report and the accompanying figure does not provide very detailed information. Therefore, we reproduced their analysis for several TCCON sites at similar latitudes for the period of our interest (September 2018 - September 2019): Karlsruhe (49.1° N), Orleans (48.0° N), Garmisch (47.5° N), Park Falls (45.9° N), Rikubetsu (43.5° N), Lamont (36.6° N), Tsukuba (36.0° N), Edwards (35.0° N), Pasadena (34.1° N), Saga (33.2° N), and Hefei (31.9° N). The results of this analysis are presented in Fig.
185 3.

We find an underestimation of the CAMS reanalysis $XCO_2$ at all TCCON sites between 30 - 50° N (except Pasadena) from October 2018 until May 2019. More specifically for Xianghe, monthly mean errors range from -2.20 ($\pm$ 1.3) ppm in January 2019 to 3.38 ($\pm$ 1.28) ppm in July 2019, which is of a similar magnitude as the bias found with WRF-GHG (where the monthly mean differences with respect to the TCCON site of Xianghe range from -2.53 ($\pm$ 1.7) ppm in December 2018 to 1.28 ($\pm$ 1.57)
ppm in July 2019).

Therefore, we assume that the error pattern detected in the $XCO_2$ time series is primarily the result of the same pattern in the background information. Moreover, this bias pattern is not found in the in situ $CO_2$ time series (Fig. 3d), likely because the relative contribution from the background to the in situ mole fractions is smaller than it is to the column data. To account for the systematic bias introduced by the background values, we apply a bias correction to the WRF-GHG simulations. Specifi-
cally, we subtract the monthly mean difference between CAMS and TCCON $XCO_2$, averaged across all TCCON sites located between 30 - 50° N (excluding Pasadena due to outlier behavior), from the model's background tracer. The resulting improvements in model performance are summarized in Table 3, and the updated time series is shown in Figure 4. After applying the correction, the mean bias error is reduced from -1.43 ($\pm$ 1.99) ppm to -0.86 ($\pm$ 1.57) ppm, with all monthly mean biases now slightly negative. As the correlation coefficient is calculated on de-seasonalized column data, it remains unaffected by the bias
correction and stays at 0.7. The RMSE shows a modest improvement, decreasing from 2.45 ppm to 1.80 ppm. The correction only has a small effect on the comparison with near-surface mole fractions, where the MBE slightly changes from -2.34 ppm to -1.65 ppm in the afternoon and from 5.82 ppm to 6.51 ppm at night. The effect on RMSE and correlation coefficients for in situ $CO_2$ are negligible.



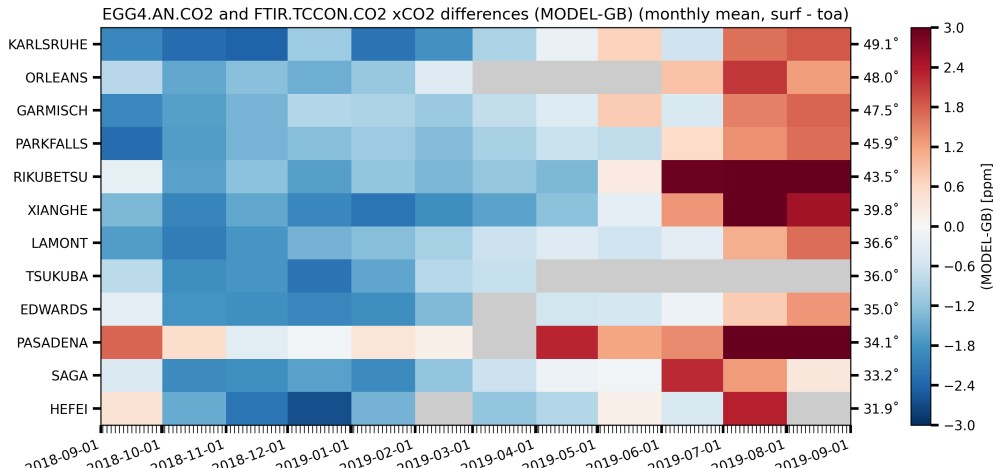

**Figure 3.** Monthly mean difference (in ppm) between CAMS reanalysis model and TCCON XCO$_2$ between 30 - 50° N over the simulation period of this study.

**Table 3.** Same as Table 2 but with bias corrected model values.

|  | insitu CO$_2$ (afternoon) | insitu CO$_2$ (night) | XCO$_2$ |
|---|---|---|---|
| MBE | -1.65 | 6.51 | -0.86 |
| RMSE | 14.49 | 25.10 | 1.80 |
| CORR | 0.75 | 0.57 | 0.70 |

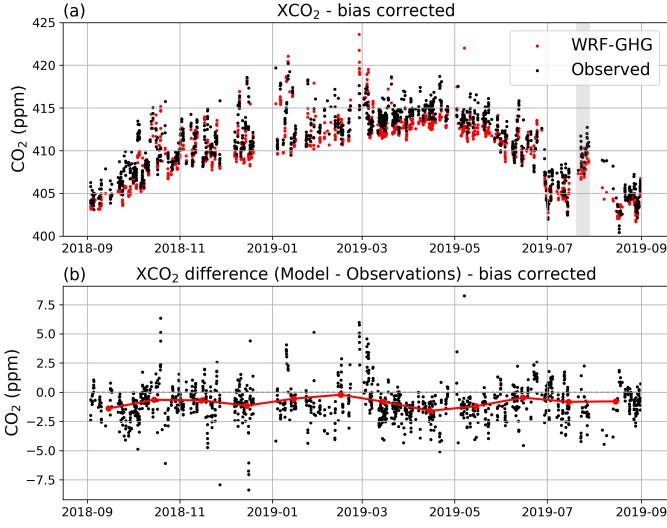

**Figure 4.** Same as Fig. 2a,c but with bias corrected model values.





**Table 4.** Statistics of the total simulated $CO_2$ mole fractions and the different tracer contributions over the complete simulation period. Q1 and Q3 represent the first and third quartile, respectively, between which 50 % of the data fall.

|  | XCO$_2$ (ppm) | | | | in situ CO$_2$ (ppm) | | | |
|---|---|---|---|---|---|---|---|---|
|  | Q1 | median | mean | Q3 | Q1 | median | mean | Q3 |
| Total | 408.32 | 412.11 | 411.37 | 414.11 | 419.44 | 430.93 | 437.86 | 450.14 |
| Background | 407.2 | 410.21 | 409.54 | 412.12 | 397.63 | 412.42 | 411.08 | 414.69 |
| Biomass burning | 0.00 | 0.00 | 0.00 | 0.00 | 0.00 | 0.00 | 0.00 | 0.00 |
| Energy | 0.36 | 0.85 | 1.07 | 1.53 | 2.74 | 6.85 | 10.51 | 14.06 |
| Residential | 0.03 | 0.06 | 0.17 | 0.17 | 0.30 | 0.65 | 1.88 | 1.95 |
| Industry | 0.24 | 0.63 | 0.79 | 1.14 | 2.70 | 5.69 | 8.53 | 10.51 |
| Transportation | 0.08 | 0.16 | 0.18 | 0.25 | 0.81 | 1.73 | 2.43 | 3.19 |
| Biosphere | -0.77 | 0.04 | -0.38 | 0.31 | -0.01 | 2.36 | 3.44 | 7.39 |
| Ocean | -0.00 | -0.00 | -0.00 | -0.00 | -0.01 | -0.00 | -0.01 | -0.00 |
| Total tracers | 0.33 | 1.23 | 1.82 | 2.80 | 7.36 | 18.99 | 26.78 | 38.30 |

## 3.2 Sector contributions to observed mole fractions

WRF-GHG tracks all fluxes in separate tracers, enabling the decomposition of the total simulated $CO_2$ mole fractions at Xianghe into contributions from different source sectors. Figure 5 shows the monthly mean values, while additionally the median and interquartile ranges are presented in Table 4. Note that all simulated hours were used for this analysis, not just the ones coinciding with observations. The main sectors contributing to the modeled $CO_2$ variability at Xianghe are energy,

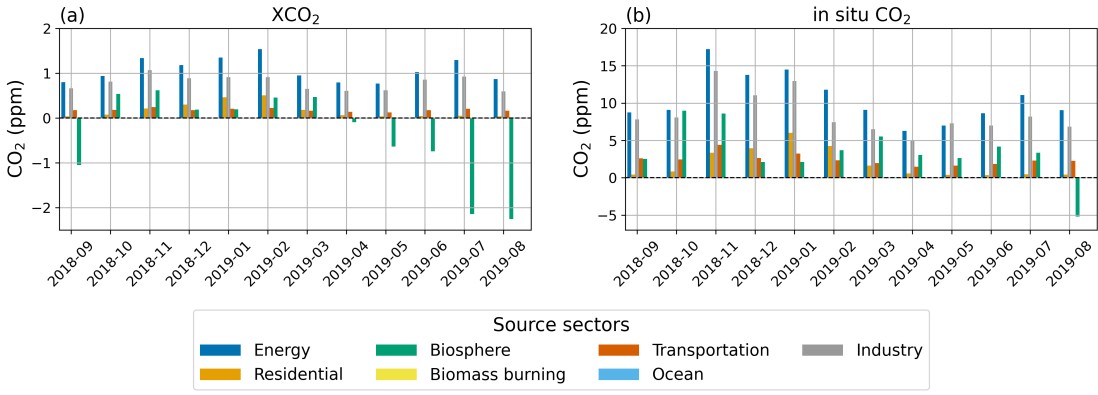

**Figure 5.** Monthly mean tracer contributions above the background for (a) XCO$_2$ and (b) in situ CO$_2$ simulated mole fractions at Xianghe.

industry, and the biosphere. For XCO$_2$, we find median values of 0.85 ppm and 0.63 ppm for the energy and industry sectors,

respectively. Furthermore, the biosphere significantly influences the column-averaged $CO_2$ values, where it acts as a sink from



April to September with a median value of -0.77 ppm during this period. During the rest of the year, the biogenic tracer acts as a small source (median value of 0.22 ppm).

Near the surface, median enhancements of in situ $CO_2$ mole fractions are 6.85 ppm and 5.69 ppm for the energy and industry sectors, respectively. The biosphere generally acts as a source throughout the year, with a median contribution of 2.69 ppm,

except in August, when it becomes a significant sink of -6.76 ppm.

The difference between the biosphere functioning as a sink in the column measurements for five months, but not in the in situ mole fractions (except in August), might result from Xianghe's location relative to strong land sinks (e.g., forests) and the different sensitivities of the measurement techniques. In situ observations are typically less sensitive to distant sources and sinks compared to FTIR observations. VPRM-computed fluxes suggest that the local biosphere near Xianghe (cropland) acts

primarily as a net source, except in August, while the forested mountains approximately 50 km north and 90 km east of the site serve as a strong sink (see Fig. 6).

Next to the three dominant sectors (biosphere, industry, and energy), transportation and also residential sources have a smaller but still relevant influence on the Xianghe data. During winter, the contribution of residential sources increases, where the highest values for the column simulations are found in February (median of 0.45 ppm) while near the surface this occurs in

January (4.28 ppm). This peak aligns with heightened residential emissions in winter, driven by increased heating demands correlated with air temperature (Guevara et al., 2021). Finally, no relevant impact was found from biomass burning and the ocean. Overall, the total tracer enhancement for the in situ mole fractions is about ten times greater than that of the column-averaged values.

### 3.3   July $XCO_2$ anomaly case study

A notable spike in $XCO_2$ levels is observed between 20-29 July (see Fig. 2a), diverging from the typical decreasing trend of $XCO_2$ from May to September, which is linked to the northern hemisphere's growing season and increased photosynthesis. We will focus on the model simulations between 7 July 2019 and 30 August 2019 to explain the causes of this $XCO_2$ summer spike, as WRF-GHG correlates well with the observations during this period (correlation coefficient of 0.84).

The total simulated $XCO_2$ increases from 405.58 ($\pm$ 1.19) ppm before to 408.90 ($\pm$ 1.19) ppm during the summer spike (20-29 July), and then decreases to 404.03 ($\pm$ 0.98) ppm after, as shown in Fig. 7a. This sudden increase of more than 3 ppm is significant and warrants further investigation. Figure 7 shows the simulated background and tracer contributions during this period. Figure 7a shows that the background $XCO_2$ remains relatively constant in July (406.84 $\pm$ 0.78 ppm), and decreases to 404.96 $\pm$ 0.73 ppm in August. It further clearly indicates an enhancement of the tracers from below the background before and

after the summer spike to above the background during the spike period. Looking at the different tracers in Fig. 7b, we see that it is mainly the biogenic tracer that has a different behavior in the spike period compared to the periods before and after. Thus, the increase in $XCO_2$ between 20-29 July is mainly linked to a less strong biogenic sink (-0.33 $\pm$ 0.57 ppm) compared to the periods before (-2.88 $\pm$ 0.71 ppm) and after (-1.76 $\pm$ 1.11 ppm).

Further analysis reveals that during the spike, a heatwave with surface temperatures up to 39°C occurred, together with 800





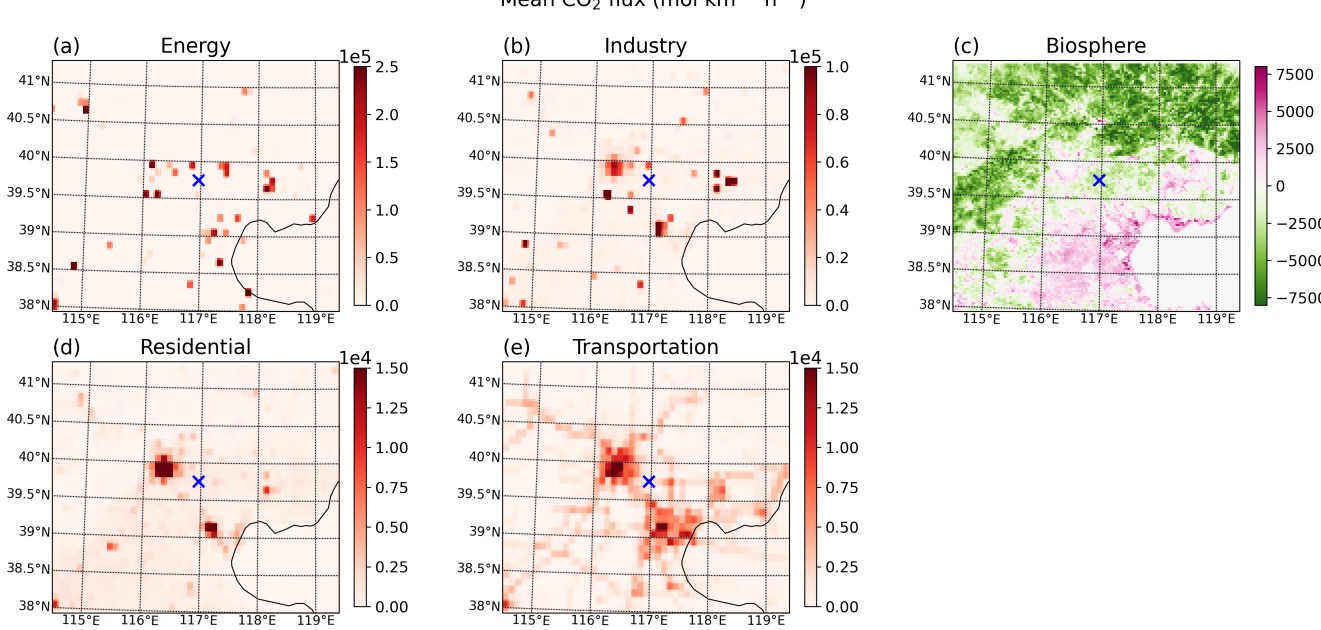

**Figure 6.** Map of the mean $CO_2$ flux (mol km$^{-2}$ h$^{-1}$) in WRF-GHG domain d03 during the entire simulation period from September 2018 until September 2019, for the most important sectors. Remark that the panels have different color scales. The location of the Xianghe site is indicated by a blue cross.

hPa winds predominantly from the west (see Fig. 7a and c). The biogenic tracer also shows increased values across a large vertical extent in the troposphere (Fig. 7c), indicating advection from other regions. Synoptic maps (Fig. A1) suggest eastward advection of a warm air mass with relatively high biogenic $CO_2$ levels, originating from the Gobi Desert and grasslands in Inner Mongolia, both areas that are characterized by sparse vegetation and elevated temperatures.

Additionally, the mean biogenic $CO_2$ flux around Xianghe, as calculated by VPRM, is slightly higher between 20 and 29 July
(average of -5 941 mol km$^{-2}$ h$^{-1}$ over domain d03) compared to the periods before and after (respectively -9 153 and -12 785 mol km$^{-2}$ h$^{-1}$). In VPRM, the respiration component is linearly dependent on surface temperature, and the gross ecosystem exchange also has a temperature dependency representing the temperature sensitivity of photosynthesis, with $CO_2$ uptake decreasing at temperatures higher than optimal (Mahadevan et al., 2008). Indeed, it has been shown that extreme temperatures impact $CO_2$ fluxes (Xu et al., 2020; Ramonet et al., 2020; Gupta et al., 2021).

Therefore, we conclude that the spike was caused by an atmospheric circulation anomaly resulting in the advection of a warm air mass with high biogenic $CO_2$ levels, along with a locally reduced ecosystem exchange due to the resulting hot temperatures.





**Figure 7.** Simulated time series of $XCO_2$ at Xianghe from 7 July 2019 to 30 August 2019, with the spike period highlighted in all panels. Daily mean (a) background tracer (cyan triangles) and total tracers (red diamonds) from WRF-GHG at Xianghe, and TCCON values (black dots). Error bars represent the standard deviation of the daily mean. Daily mean 800 hPa wind direction is indicated by wind barbs at the bottom. (b) Time series of different tracer contributions at Xianghe ,with hourly values shown as thin lines and points for TCCON observation times. (c) Color coded vertical profiles of the biogenic $CO_2$ contributions (left y-axis) shown in red and blue, and surface temperature (right y-axis) in black.



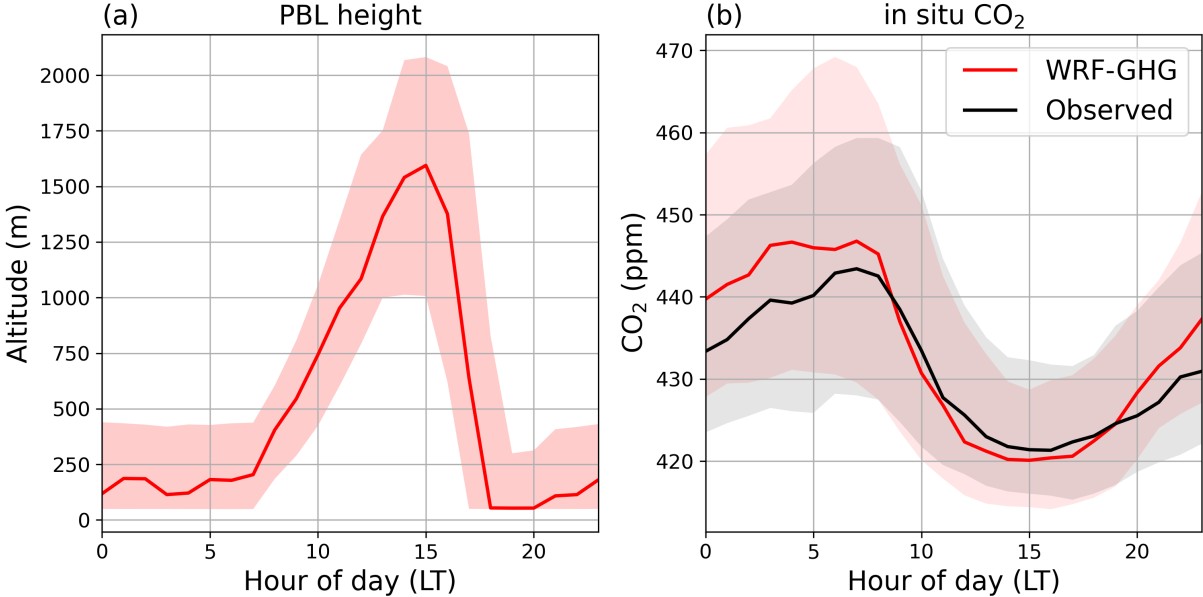

**Figure 8.** Hourly median and interquartile range of the (a) simulated planetary boundary layer height, and observed and simulated surface (b) $CO_2$ mole fraction at Xianghe.

## 3.4 Diurnal cycle analysis of in situ data

The planetary boundary layer (PBL) plays a crucial role in regulating near-surface $CO_2$ mole fractions. Figure 8 displays the diurnal variation of the PBL height as simulated by WRF-GHG, along with the corresponding $CO_2$ mole fractions near the
surface (both simulated and observed).

During the day, solar radiation promotes turbulent mixing, leading to a deepening of the PBL and the dilution of near-surface $CO_2$. The PBL reaches its maximum height around 15:00 local time (LT), coinciding with the lowest simulated surface $CO_2$ mole fractions, with median (and interquartile) values of 420.10 (414.41 - 429.69) ppm, though mole fractions remain comparably low from 14:00 to 17:00. Conversely, during the night, radiative cooling leads to the formation of a stable, shallow PBL,
trapping $CO_2$ near the surface and causing mole fractions to rise. The peak $CO_2$ mole fractions in WRF-GHG are reached at 7:00 LT, with values of 446.78 (429.57 - 467.64) ppm, but are relatively stable between 3:00 and 8:00 LT (median values between 445.21 and 446.26 ppm). As the sun rises and the PBL height begins to increase again, the $CO_2$ mole fractions drop, giving rise to a characteristic diurnal cycle.

WRF-GHG successfully captures the general shape of this diurnal cycle, but discrepancies remain in the amplitude and timing.
The model slightly underestimates daytime mole fractions, the lowest observed values occur between 14:00 and 16:00 LT, with the minimum at 16:00 of 421.32 (415.80 – 431.76) ppm. During nighttime, the model tends to overestimate the $CO_2$ build-up, with observed mole fractions showing a distinct peak at 7:00, reaching 443.42 (428.00–459.32) ppm. This results in an overestimation of the diurnal amplitude by approximately 4.58 ppm. Such a nighttime overestimation was not observed for




$CH_4$ at the same site (Callewaert et al., 2024), suggesting that the bias is more likely related to the surface fluxes of $CO_2$ than to PBL dynamics.

Unlike $CH_4$, the near-surface diurnal cycle of $CO_2$ is heavily influenced by photosynthetic uptake and respiratory release from vegetation. This creates a strong correlation between solar-driven PBL dynamics and biogenic $CO_2$ fluxes, a phenomenon known as the atmospheric $CO_2$ *rectifier effect* (Larson and Volkmer, 2008). During the day, vegetation takes up $CO_2$ through photosynthesis, contributing to lower mole fractions, while at night, respiration dominates, releasing $CO_2$ back into the atmosphere. Consequently, inaccuracies in the biogenic fluxes can amplify or dampen the PBL-driven signal.

## 3.5 Sensitivity experiments

Several sources of uncertainty may affect the accuracy of the simulated anthropogenic and biogenic $CO_2$ fluxes in WRF-GHG. First of all, the parameters used in VPRM in this study are based on Li et al. (2020), who optimized them for ecosystems in the United States. Applying these values to China likely introduces regional mismatches. Moreover, the linear formulation of the respiration term in VPRM has been identified as a source of potential bias (Dong et al., 2021; Hu et al., 2021). A third concern is the land cover classification. VPRM uses the SYNMAP product (Jung et al., 2006), which is a 1-km global land cover map that classifies the area around Xianghe as 100% cropland. While broadly consistent with the regional land use, this dataset does not account for increasing urbanization during the last decades. In WRF-GHG, built-up areas are assigned zero biogenic flux, so their omission could contribute to the observed nighttime overestimation of respiration and daytime photosynthetic uptake. Further, the representation of anthropogenic emission heights may also affect the modeled surface mole fractions. In this study, all anthropogenic $CO_2$ emissions are released in the lowest model layer, which simplifies reality. Especially for sectors such as energy and industry, this is a crude approximation, since facilities like power plants typically emit at elevated stacks. Previous work by Brunner et al. (2019) has shown that ignoring the vertical distribution of emissions can lead to overestimation of near-surface mole fractions.

Finally, while it is well known that uncertainties in simulating planetary boundary layer (PBL) dynamics can substantially affect near-surface $CO_2$ mole fractions, the influence of different PBL parameterization schemes is not explored in the current sensitivity experiments. WRF-GHG offers several PBL schemes, some of which were tested and discussed in Part 1 of this work (Callewaert et al., 2024). Here, the focus is instead on evaluating how model configuration choices related to $CO_2$ fluxes impact the simulated mole fractions.

### 3.5.1 Emission height sensitivity

To evaluate the impact of emission injection height on simulated $CO_2$ mole fractions at Xianghe, we compare the SFC and PROF configurations of the BASE sensitivity experiment (see Sect. 2.3). Figure 9 presents the median diurnal cycle of the anthropogenic $CO_2$ tracer across the four 14-day simulation periods (top), together with the total simulated and observed $CO_2$ mole fractions (bottom). Similarly, Table 5 provides a summary of the impact on the anthropogenic tracer. Results for the LC and PARAM experiments are similar to those of BASE and are therefore not discussed further here.

Simulations using elevated emission profiles (PROF) consistently yield lower near-surface $CO_2$ mole fractions than those





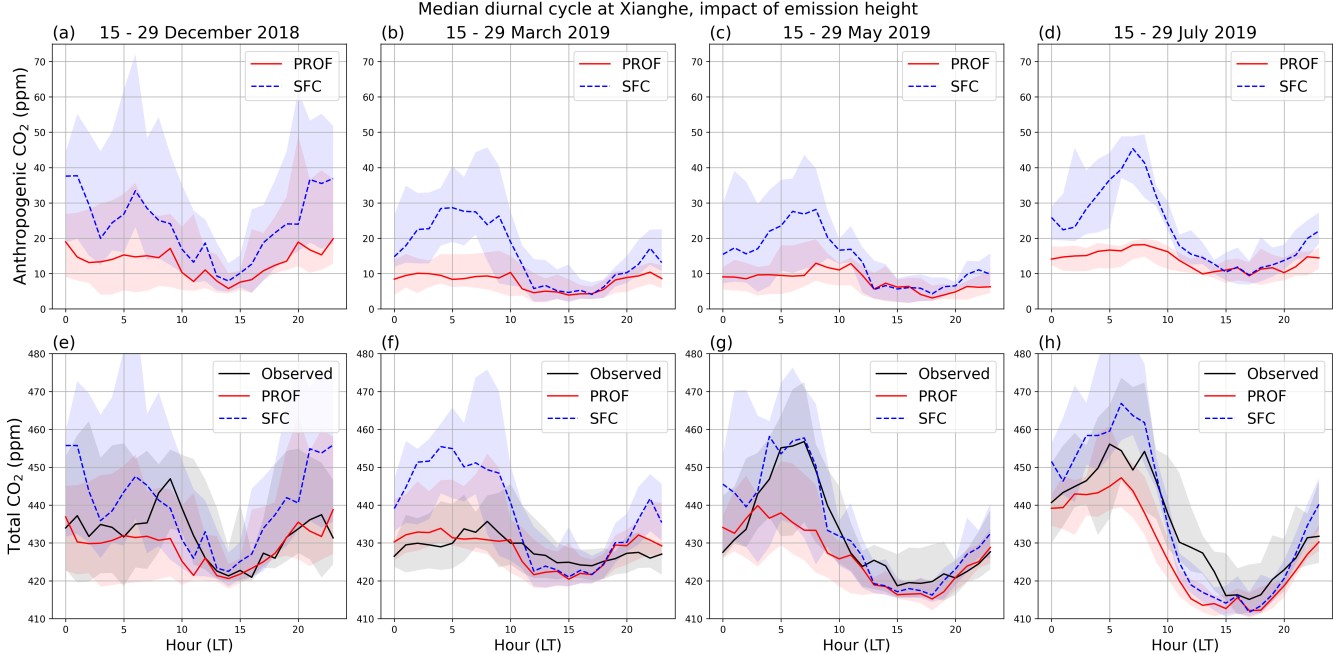

**Figure 9.** Median diurnal cycle of in situ $CO_2$ mole fractions (ppm) at Xianghe. The solid red line presents the simulated values of the BASE sensitivity experiment using vertical profiles for the anthropogenic emissions (PROF), while the dashed blue line represents the simulated $CO_2$ values using only surface emissions (SFC). Observations are plotted in black.

**Table 5.** Mean and standard deviation (in ppm) of the total anthropogenic $CO_2$ tracer contribution (sum of industry, energy, transportation and residential tracer) to near-surface mole fractions at Xianghe for BASE sensitivity experiment, split by surface (SFC) and profile (PROF) emissions, and simulation period. 'All' indicates that all simulated hours (0 - 23 LT) were used to calculate the metrics, in contrast to 'Afternoon' (13 - 18 LT) and 'Night' (3 - 8 LT).

|  |  | December 2018 | March 2019 | May 2019 | July 2019 |
|---|---|---|---|---|---|
| All | SFC | $34.32 \pm 33.64$ | $18.94 \pm 18.74$ | $15.59 \pm 14.35$ | $24.25 \pm 14.78$ |
|  | PROF | $20.37 \pm 20.47$ | $9.51 \pm 7.76$ | $8.5 \pm 6.16$ | $13.98 \pm 5.32$ |
| Afternoon | SFC | $21.23 \pm 26.51$ | $8.5 \pm 8.5$ | $6.9 \pm 5.89$ | $14.09 \pm 9.72$ |
|  | PROF | $15.91 \pm 19.76$ | $6.67 \pm 5.82$ | $5.92 \pm 4.7$ | $11.8 \pm 5.3$ |
| Night | SFC | $40.25 \pm 34.99$ | $30.08 \pm 20.25$ | $24.73 \pm 16.96$ | $37.67 \pm 15.28$ |
|  | PROF | $20.28 \pm 17.76$ | $11.86 \pm 8.93$ | $11.16 \pm 7.45$ | $15.92 \pm 4.72$ |





**Table 6.** Overview of statistical metrics (MBE, RMSE and CORR) for BASE sensitivity experiment, split by surface (SFC) and profile (PROF) emissions, and simulation period. Values for MBE and RMSE are given in ppm.

| | | MBE | | | | RMSE | | | | CORR | | | |
|---|---|---|---|---|---|---|---|---|---|---|---|---|---|
| | | Dec | Mar | May | Jul | Dec | Mar | May | Jul | Dec | Mar | May | Jul |
| All | SFC | 10.04 | 10.8 | 0.69 | 1.99 | 31.9 | 22.41 | 17.22 | 19.22 | 0.55 | 0.47 | 0.61 | 0.64 |
| | PROF | -3.91 | 1.38 | -6.4 | -8.27 | 23.51 | 11.47 | 16.08 | 16.82 | 0.53 | 0.49 | 0.62 | 0.67 |
| Afternoon | SFC | 3.66 | -0.31 | -3.88 | -3.52 | 26.83 | 11.24 | 10.04 | 15.74 | 0.43 | 0.50 | 0.28 | 0.02 |
| | PROF | -1.65 | -2.13 | -4.86 | -5.81 | 22.69 | 9.81 | 10.32 | 13.73 | 0.40 | 0.53 | 0.24 | 0.04 |
| Night | SFC | 13.44 | 21.45 | 1.02 | 11.35 | 30.9 | 29.75 | 22.04 | 25.64 | 0.66 | 0.43 | 0.52 | 0.35 |
| | PROF | -6.54 | 3.23 | -12.55 | -10.4 | 19.35 | 14.0 | 23.61 | 20.77 | 0.75 | 0.32 | 0.48 | 0.42 |

with surface-only emissions (SFC), with the most pronounced differences occurring during nighttime. The largest reduction is observed in July, where mean nighttime mole fractions in the PROF simulation are 21.75 ppm lower than in the SFC case. These findings align with the results of Brunner et al. (2019), who highlighted the critical role of emission height in determining

near-surface $CO_2$ mole fractions, particularly under weak mixing conditions. The substantial nighttime differences are likely driven by the proximity of strong point sources to the Xianghe site (Fig. 6a,b), and whether these emissions are released below or above the shallow nocturnal PBL.

An overview of the key statistical performance metrics with respect to the observations at Xianghe is given in Table 6. Applying vertically distributed emissions to the WRF-GHG simulations at Xianghe shifts the MBE at night from a positive to a negative

value. While the absolute magnitude of the MBE decreases significantly at night, it increases only slightly in the afternoon. Additionally, the use of elevated emissions leads to a reduction in the RMSE compared to the surface-only configuration, while the correlation coefficient remains largely unaffected. The use of elevated emission profiles also has a pronounced effect on the simulated diurnal amplitude of near-surface $CO_2$. Compared to the surface-only configuration, which strongly overestimates the amplitude, the more realistic vertical distribution results in a better agreement with observations—particularly in March and

July. In March, for example, the amplitude overestimation is reduced from 22.73 ppm to just 1.74 ppm, and in July from 14.16 ppm to -5.96 ppm. These results illustrate the strong influence of anthropogenic emission height on simulated near-surface $CO_2$ mole fractions at Xianghe. Although some discrepancies with observations persist, the application of a more realistic vertical emission distribution clearly improves model performance.

The implementation of elevated anthropogenic emissions has a minimal effect on the column-averaged $XCO_2$ mole fractions

at Xianghe, see statistical metrics in Table A2.

### 3.5.2 Biogenic flux and land cover sensitivity

To evaluate the impact of land cover representation on the VPRM-calculated $CO_2$ fluxes, we compare the BASE and LC sensitivity experiments. All figures and values below use the vertically distributed anthropogenic fluxes (PROF). In the BASE simulation, the WRF-GHG grid cell containing the Xianghe site is classified as 100% cropland using the SYNMAP dataset. In





**Figure 10.** Median diurnal cycle of biogenic $CO_2$ flux (top), biogenic $CO_2$ tracer contribution (middle) to the near surface $CO_2$ mole fractions (bottom) at Xianghe for the different simulation periods (columns). Different curves (colors) represent different sensitivity experiments BASE, LC and PARAM. Observations are plotted in black in the bottom row.

contrast, the Copernicus Dynamic Land Cover data, used in the LC and PARAM simulations, classifies the same cell as 68.88% cropland, 23.5% no vegetation (representing urban surfaces, water, ice, rocks, etc. ), 3.45% mixed forest, 2.03% wetland, 1.5% shrubland, and 0.64% grassland. A comparison of the two land cover datasets over the innermost WRF-GHG domain (d03) is shown in Figure A3. Due to its higher spatial resolution and more up-to-date information, the Copernicus dataset introduces more heterogeneous vegetation fractions and especially reduces the cropland fraction while increasing the urban land category compared to SYNMAP.

The top panels of Figure 10 present the median diurnal cycles of the biogenic $CO_2$ flux, expressed as NEE, at Xianghe across the four 14-day simulation periods. As expected, differences between the experiments are negligible in winter months (Decem-





**Table 7.** Mean and standard deviation of the biogenic $CO_2$ tracer at Xianghe for the different sensitivity experiments BASE, LC and PARAM, per simulation period. 'All' indicates all simulated hours (0 - 23 LT) were used to calculate the metrics, in contrast to 'Afternoon' (13 - 18 LT) and 'Night' (3 - 8 LT).

|  |  | December 2018 | March 2019 | May 2019 | July 2019 |
|---|---|---|---|---|---|
|  | BASE | $2.29 \pm 2.67$ | $6.25 \pm 5.6$ | $3.79 \pm 9.35$ | $8.07 \pm 12.89$ |
| All | LC | $2.28 \pm 2.66$ | $4.67 \pm 4.19$ | $2.15 \pm 7.91$ | $6.62 \pm 10.04$ |
|  | PARAM | $4.38 \pm 4.91$ | $9.49 \pm 7.51$ | $10.53 \pm 10.09$ | $17.24 \pm 13.85$ |
|  | BASE | $1.51 \pm 1.62$ | $2.99 \pm 3.96$ | $-3.13 \pm 3.46$ | $-3.4 \pm 4.0$ |
| Afternoon | LC | $1.5 \pm 1.6$ | $2.35 \pm 3.06$ | $-3.02 \pm 4.48$ | $-2.02 \pm 3.28$ |
|  | PARAM | $2.4 \pm 2.58$ | $5.77 \pm 6.26$ | $3.08 \pm 2.63$ | $5.35 \pm 5.33$ |
|  | BASE | $2.46 \pm 2.96$ | $8.87 \pm 6.27$ | $11.21 \pm 10.85$ | $22.16 \pm 11.91$ |
| Night | LC | $2.44 \pm 2.95$ | $6.54 \pm 4.73$ | $7.74 \pm 9.28$ | $17.39 \pm 9.67$ |
|  | PARAM | $5.14 \pm 5.66$ | $12.75 \pm 8.05$ | $18.56 \pm 12.51$ | $32.2 \pm 13.25$ |

ber and March) due to minimal biospheric activity. In contrast, during May and July, the LC simulation exhibits both reduced daytime $CO_2$ uptake and lower nighttime respiration compared to BASE. This weakening of the biospheric signal is consistent

with the increased fraction of non-vegetated land in LC, which suppresses VPRM-driven fluxes.

This change in biogenic flux is reflected in the simulated biogenic $CO_2$ tracer at Xianghe (Fig.10e–h, Table 7). However, we notice that the magnitude of the difference between LC and BASE is more pronounced at night. For instance, in July, the mean ($\pm$ standard deviation) difference in the biogenic tracer is –4.78 ($\pm$ 2.73) ppm during nighttime, while the daytime difference is only –1.38 ($\pm$ 1.19) ppm.

To assess the impact of the VPRM parameterization on $CO_2$ simulations at Xianghe, we compare the LC and PARAM experiments. Both simulations use the 100-m Copernicus Dynamic Land Cover dataset but differ in their VPRM parameter tables (see Table 1). The comparison reveals that PARAM systematically produces more positive biogenic $CO_2$ fluxes than LC, both during daytime and nighttime. This difference can be attributed to higher values of the parameters $\alpha$ and $\beta$ in the Glauch parameter set (Table A1), which enhance respiration rates. In contrast, the gross ecosystem exchange (GEE) is comparable between

the two configurations, indicating that the net effect is primarily driven by increased respiration rather than photosynthetic uptake. The change in biogenic flux is reflected in the biogenic $CO_2$ tracer mole fractions at Xianghe. In July, for example, tracer values in PARAM are on average 10.61 ($\pm$ 5.03) ppm higher than in LC. This difference is again more pronounced at night (14.82 $\pm$ 5.27ppm) than during the afternoon (7.37 $\pm$ 3.48 ppm).

A summary of the model performance of the different experiments using vertically distributed anthropogenic emissions

(PROF) is shown in Table 8. Generally, when anthropogenic emissions are vertically distributed, the PARAM experiment yields the best agreement with observations. Across all months, PARAM shows the highest correlation coefficients and the lowest MBE and RMSE, with the exception of March. In that month, the LC experiment slightly outperforms PARAM, with a smaller MBE (–0.2 ppm vs. 4.61 ppm) and RMSE (10.73 ppm vs. 13.09 ppm).





**Table 8.** Overview of statistical metrics for the different sensitivity experiments BASE, LC and PARAM with respect to the observations, per simulation period. The total simulated $CO_2$ mole fractions use the vertically distributed anthropogenic fluxes (PROF). Values for MBE and RMSE are given in ppm.

|  |  | MBE | | | | RMSE | | | | CORR | | | |
|---|---|---|---|---|---|---|---|---|---|---|---|---|---|
|  |  | Dec | Mar | May | Jul | Dec | Mar | May | Jul | Dec | Mar | May | Jul |
|  | BASE | -3.91 | 1.38 | -6.4 | -8.27 | 23.51 | 11.47 | 16.08 | 16.82 | 0.53 | 0.49 | 0.62 | 0.67 |
| All | LC | -3.92 | -0.2 | -8.04 | -9.72 | 23.52 | 10.73 | 17.2 | 17.3 | 0.53 | 0.48 | 0.58 | 0.69 |
|  | PARAM | -1.82 | 4.61 | 0.34 | 0.82 | 23.6 | 13.09 | 13.79 | 13.64 | 0.55 | 0.5 | 0.68 | 0.73 |
|  | BASE | -1.65 | -2.13 | -4.86 | -5.81 | 22.69 | 9.81 | 10.32 | 13.73 | 0.4 | 0.53 | 0.24 | 0.04 |
| Afternoon | LC | -1.67 | -2.78 | -4.75 | -4.43 | 22.69 | 9.7 | 10.46 | 12.79 | 0.4 | 0.53 | 0.23 | 0.1 |
|  | PARAM | -0.76 | 0.64 | 1.35 | 2.88 | 22.88 | 10.63 | 9.08 | 12.93 | 0.41 | 0.52 | 0.28 | 0.19 |
|  | BASE | -6.54 | 3.23 | -12.55 | -10.4 | 19.35 | 14.0 | 23.61 | 20.77 | 0.75 | 0.32 | 0.48 | 0.42 |
| Night | LC | -6.55 | 0.9 | -16.02 | -15.18 | 19.37 | 12.9 | 25.67 | 22.87 | 0.75 | 0.31 | 0.45 | 0.47 |
|  | PARAM | -3.86 | 7.11 | -5.2 | -0.44 | 19.01 | 16.1 | 18.76 | 16.57 | 0.74 | 0.34 | 0.61 | 0.54 |

The various VPRM inputs have only a minor influence on the column-averaged $XCO_2$ mole fractions at Xianghe, as indicated by the statistical metrics in Table A4.

### 3.5.3 Discussion

In the one-year simulation, we observed an overestimation of near-surface $CO_2$ diurnal amplitude at Xianghe, mainly due to overestimated nighttime values, and a smaller underestimation during the day. To better understand and address these discrepancies, several targeted sensitivity experiments were performed. These experiments provide valuable insights into the influence of key model settings and help guide future improvements for simulating $CO_2$ at this site and elsewhere.

We find that both applying vertical profiles to anthropogenic emissions and using the more recent Copernicus land cover map systematically reduce nighttime $CO_2$ mole fractions at Xianghe, leading to an improved representation of reality in the model. In addition, the experiments clearly demonstrate that alternative VPRM parameter values can significantly impact simulated $CO_2$ mole fractions at Xianghe. These findings highlight the sensitivity of modeled results to choices in land cover classification and biogenic flux parameterization, underscoring the importance of careful model configuration.

That said, the scope of our experiments is limited. Only four two-week periods were tested due to computational constraints. Ideally, a full-year sensitivity analysis would better capture variability under different meteorological conditions, but this was not feasible in the current study. This limitation introduces some ambiguity in interpreting the results. For example, in the May sensitivity tests, the surface-emission (SFC) simulation aligned more closely with observations than the elevated-emission (PROF) run, despite this month being selected due to poor agreement with near-surface mole fractions in the one-year BASE simulation. This apparent contradiction reflects the fact that the model–observation mismatch occurred mainly in the first half of May, which was not included in the shorter simulations. Moreover, the poorer performance of PROF in the second half of



May remains unexplained, possibly pointing to remaining inaccuracies in the vertical emission profiles or to meteorological factors and uncertainties in modeling of PBL dynamics.

Additional limitations should be noted. Currently, no standard VPRM parameter set specifically developed for China exists. The only known attempt to tailor VPRM parameters to Chinese conditions is the study by Dayalu et al. (2018), which subdivides croplands into seasonally varying subtypes. However, due to its complexity, the lack of readily available seasonal land cover input data, and recent work by Seo et al. (2024) that indicates that the parameter values used in our study (from Li et al. (2020)) perform better than those from Dayalu et al. (2018) over East Asia, we did not adopt the latter parameterization in

the current study. Finally, the simplified formulation of the respiration term in VPRM was not modified, as this would require additional parameters that are not currently available for China and would introduce further uncertainty.

Despite these limitations, our experiments demonstrate that specific adjustments—such as adopting the Copernicus land cover map and applying vertical emission profiles—can substantially improve model performance. While the combination of Copernicus land cover and VPRM parameters from Glauch et al. (2025) yielded the best match with observations for simulations

using vertical emission profiles, this was not the case when surface emissions were used (A3). In the latter configuration, the LC experiment performed better than PARAM.

Overall, we recommend the use of the Copernicus land cover map (or a comparable up-to-date high-resolution dataset) for its more realistic surface representation and advise the application of vertical profiles for anthropogenic emissions. However, more research is needed to assess whether the Brunner et al. (2019) profiles are suitable for China and whether they align with

the spatial and sectoral characteristics of the used emissions inventory. Finally, we emphasize the need for a dedicated study to optimize VPRM parameters in this region, ideally based on eddy-covariance measurements from multiple tower sites. These efforts will contribute to more accurate simulations of biogenic and anthropogenic $CO_2$ fluxes and improved atmospheric $CO_2$ modeling at Xianghe and across East Asia.

## 4   Conclusions

This study is the second part of a broader investigation into greenhouse gas variability and model performance at the Xianghe site, following earlier work focused on $CH_4$. Here, we shift the focus to $CO_2$, aiming to better understand the observed variability through source attribution and model simulations. Using the WRF-GHG model, we performed a one-year simulation of both surface and column-averaged $CO_2$, evaluated model performance against FTIR and in situ observations, and carried out sensitivity experiments to assess the impact of key model settings.

Model evaluation against FTIR observations at Xianghe shows that WRF-GHG is capable of capturing the temporal variability in column-averaged $CO_2$ ($XCO_2$), with a correlation coefficient of 0.7. However, a systematic bias was identified in the model's background $CO_2$ values from CAMS, with a negative offset exceeding 2 ppm between September and May. After applying a bias correction based on monthly mean CAMS–TCCON differences, the mean bias error was reduced to -0.86 ppm. These findings underscore the importance of accurate boundary conditions when simulating $XCO_2$, particularly due to the long

atmospheric lifetime of $CO_2$ and the relatively small contribution of regional emissions to the total column. In our simulations,



emissions within the model domain contributed only $\sim$ 1.82 ppm to $XCO_2$, making the column signal highly sensitive to background mole fractions. Furthermore, the model successfully captured a strong positive anomaly observed in July 2019, attributed to the advection of a warm, $CO_2$-rich air mass. This case study illustrates the value of combining transport and mole fraction diagnostics for interpreting episodic events in column data and highlights the dominant role of synoptic meteorology
in driving short-term variability in $XCO_2$.

For near-surface $CO_2$ mole fractions, WRF-GHG shows good agreement with afternoon observations at Xianghe, achieving a correlation coefficient of 0.75 and a mean bias error of –1.65 ppm after bias correction. In contrast to $XCO_2$, near-surface $CO_2$ was more strongly influenced by local sources, with a mean tracer enhancement of 26.78 ppm, thereby reducing the relative importance of boundary condition errors. Nighttime mole fractions are consistently overestimated, with a mean bias of 6.51
ppm and a lower correlation of 0.57. These discrepancies are reflected in the diurnal cycle: while the model captures the overall structure driven by planetary boundary layer dynamics, it overestimates the daily amplitude of 22.1 ppm by 4.58 ppm. Likely causes include inaccuracies in biogenic $CO_2$ fluxes and the vertical distribution of anthropogenic emissions.

Additional sensitivity experiments show that applying vertical emission profiles and a more recent land cover map can reduce nighttime $CO_2$ mole fractions and improve agreement with observations. Adjustments to the VPRM vegetation parameters
substantially affected near-surface mole fractions, underscoring the critical role of appropriate parameter selection—especially in the absence of a standardized VPRM configuration for China.

Tracer analysis confirms that the industry and energy sectors are the dominant contributors to $CO_2$ levels at Xianghe, while the biosphere plays a secondary, seasonal role. For $XCO_2$, the biosphere acts as a sink from April to September (–0.77 ppm on average) and a weak source in the remaining months (+0.22 ppm). At the surface, biospheric uptake is only seen in August
(–6.76 ppm), while emissions dominate the rest of the year (+2.69 ppm on average). These differences illustrate the greater local sensitivity of in situ measurements compared to column observations and the varying spatial influence of different source types.

Overall, this study demonstrates the value of using a modeling framework like WRF-GHG to interpret both temporal and sectoral variations in surface and column $CO_2$ observations. It also highlights that model accuracy is strongly dependent on
appropriate configuration choices, including the representation of boundary conditions, vertical emission profiles, and biogenic flux parameterizations. Addressing these factors is essential for improving simulations and supporting more accurate source attribution of observed $CO_2$ variability.

*Code and data availability.* The ERA5 and CAMS reanalysis data set (Hersbach et al., 2023a, b), used as input for the WRF-GHG simulations, was downloaded from the Copernicus Climate Change Service (C3S) Climate Data Store (2022). The CAMS-GLOB-ANT v5.3
emissions (Granier et al., 2019; Soulie et al., 2023) and temporal profiles CAMS-GLOB-TEMPO v3.1 (Guevara et al., 2021) are archived and distributed through the Emissions of atmospheric Compounds and Compilation of Ancillary Data (ECCAD) platform. The WRF-Chem model code is distributed by NCAR (https://doi.org/10.5065/D6MK6B4K, NCAR, 2020). The WRF-GHG simulation output created in the context of this study can be accessed on https://doi.org/10.18758/P34WJEW2 (Callewaert, 2023). The TCCON data were obtained from the





**Table A1.** VPRM parameter values for different vegetation classes.

|  |  | Evergreen forest | Deciduous forest | Mixed forest | Shrubland | Savanna | Cropland | Grassland | Wetland |
|---|---|---|---|---|---|---|---|---|---|
| Li table | $PAR_0$ | 745.306 | 514.13 | 419.5 | 590.7 | 600 | 1074.9 | 717.1 | 392.666 |
|  | $\lambda$ | 0.13 | 0.1 | 0.1 | 0.18 | 0.18 | 0.085 | 0.115 | 0.1377 |
|  | $\alpha$ | 0.1247 | 0.092 | 0.2 | 0.0634 | 0.2 | 0.13 | 0.0515 | 0.0779 |
|  | $\beta$ | 0.2496 | 0.8430 | 0.27248 | 0.2684 | 0.3376 | 0.542 | -0.0986 | 0.0902 |
| Glauch table | $PAR_0$ | 521.9 | 500.8 | 451.1 | 444.1 |  | 960.8 | 443.4 | 399.7 |
|  | $\lambda$ | 0.13 | 0.13 | 0.14 | 0.1 |  | 0.09 | 0.22 | 0.12 |
|  | $\alpha$ | 0.21 | 0.23 | 0.19 | 0.08 |  | 0.17 | 0.27 | 0.3 |
|  | $\beta$ | 1.15 | 1.26 | 0.93 | 0.56 |  | 1.14 | 1.63 | -0.39 |

**Table A2.** Same as Table 6 but for $XCO_2$.

|  |  | MBE | | | | RMSE | | | | CORR | | | |
|---|---|---|---|---|---|---|---|---|---|---|---|---|---|
|  |  | Dec | Mar | May | Jul | Dec | Mar | May | Jul | Dec | Mar | May | Jul |
| All | SFC | -3.15 | -2.15 | -1.45 | 0.37 | 4.07 | 2.36 | 1.8 | 1.36 | 0.6 | 0.67 | 0.38 | 0.59 |
|  | PROF | -3.55 | -2.22 | -1.56 | 0.02 | 4.29 | 2.41 | 1.89 | 1.09 | 0.63 | 0.69 | 0.35 | 0.71 |

TCCON Data Archive hosted by CaltechDATA at https://tccondata.org (Zhou et al., 2022), while the surface observations at Xianghe were

received through private communication with the co-authors.

**Appendix A: Additional tables and figures**

*Author contributions.* SC made the model simulations and performed the formal analysis, investigation and visualization. The research was conceptualized by SC, MDM and EM and supervised by MDM and EM. MZ, TW and PW have provided the observational in situ data at Xianghe. BL supported with computing tools to correctly compare the model with TCCON data. SC prepared the initial draft of this

manuscript while it was reviewed and edited by MZ, BL, TW, MDM, EM and PW.

*Competing interests.* The contact authors have declared that neither they nor their co-authors have any competing interests.

*Disclaimer.* The results contain modified Copernicus Climate Change Service information 2022. Neither the European Commission nor ECMWF is responsible for any use that may be made of the Copernicus information or data it contains.





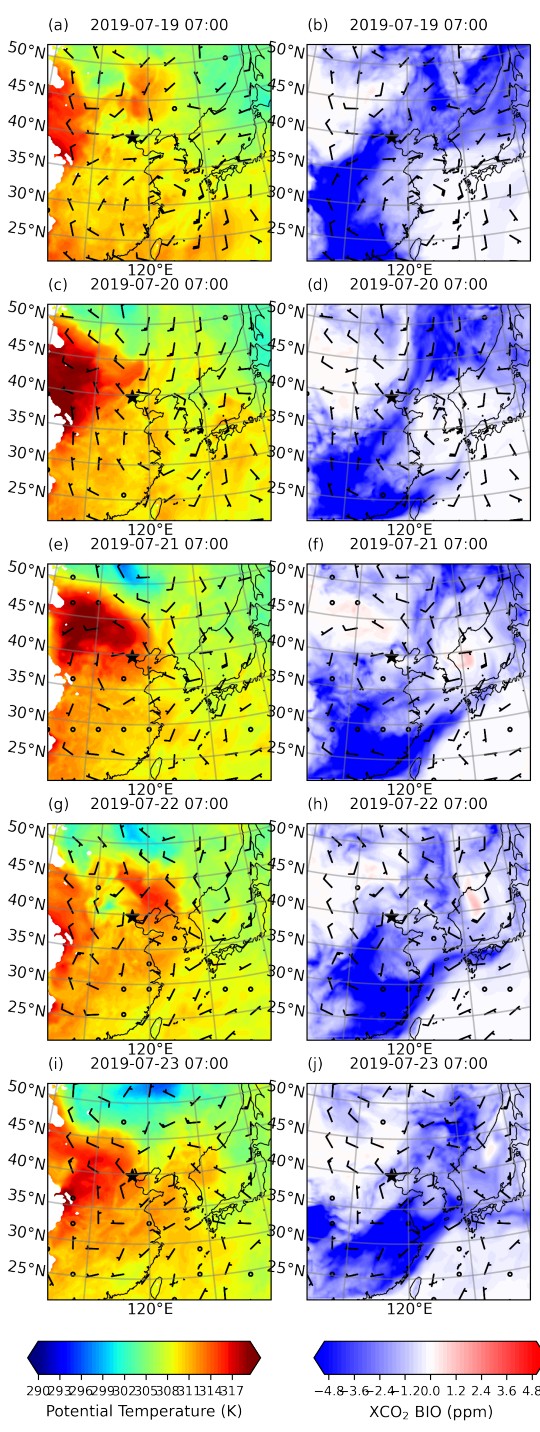

**Figure A1.** Maps of eastern China at 800 hPa for 19–23 July 2019, 07:00 UTC (15:00 LT). Panels in the first column ((a) - (i)) show potential temperature (K, in color) with wind barbs at 800 hPa; panels in the second column ((b) - (j)) show biogenic $XCO_2$ enhancements (ppm, in color) with the same wind vectors. The Xianghe site is marked by a black star.





**Table A3.** Same as Table 8 but with SFC anthropogenic emissions instead of PROF.

|  |  | MBE | | | | RMSE | | | | CORR | | | |
|---|---|---|---|---|---|---|---|---|---|---|---|---|---|
|  |  | Dec | Mar | May | Jul | Dec | Mar | May | Jul | Dec | Mar | May | Jul |
| All | BASE | 10.04 | 10.8 | 0.69 | 1.99 | 31.9 | 22.41 | 17.22 | 19.22 | 0.55 | 0.47 | 0.61 | 0.64 |
|  | LC | 10.03 | 9.22 | -0.95 | 0.55 | 31.9 | 20.95 | 16.91 | 17.69 | 0.55 | 0.46 | 0.59 | 0.64 |
|  | PARAM | 12.13 | 14.04 | 7.43 | 11.1 | 33.76 | 24.78 | 18.57 | 21.67 | 0.56 | 0.49 | 0.65 | 0.69 |
| Afternoon | BASE | 3.66 | -0.31 | -3.88 | -3.52 | 26.83 | 11.24 | 10.04 | 15.74 | 0.43 | 0.5 | 0.28 | 0.02 |
|  | LC | 3.65 | -0.95 | -3.76 | -2.14 | 26.82 | 10.81 | 10.21 | 14.99 | 0.43 | 0.5 | 0.27 | 0.06 |
|  | PARAM | 4.55 | 2.47 | 2.33 | 5.16 | 27.4 | 12.82 | 9.62 | 16.69 | 0.43 | 0.5 | 0.3 | 0.13 |
| Night | BASE | 13.44 | 21.45 | 1.02 | 11.35 | 30.9 | 29.75 | 22.04 | 25.64 | 0.66 | 0.43 | 0.52 | 0.35 |
|  | LC | 13.42 | 19.12 | -2.45 | 6.57 | 30.9 | 27.68 | 21.63 | 22.94 | 0.66 | 0.42 | 0.5 | 0.36 |
|  | PARAM | 16.12 | 25.32 | 8.37 | 21.33 | 33.63 | 32.97 | 22.77 | 30.52 | 0.66 | 0.44 | 0.59 | 0.44 |

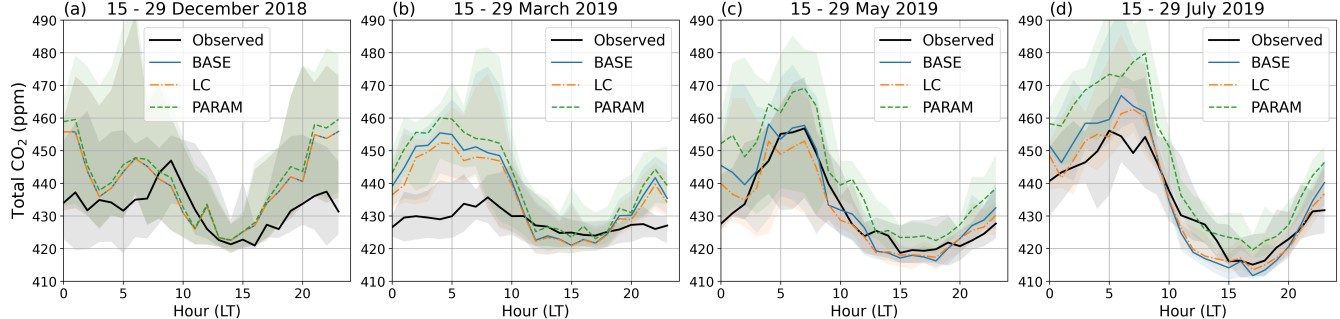

**Figure A2.** Same as bottom panels of Fig. 10 but with SFC anthropogenic emissions instead of PROF.

**Table A4.** Same as Table 8 but for $XCO_2$.

|  |  | MBE | | | | RMSE | | | | CORR | | | |
|---|---|---|---|---|---|---|---|---|---|---|---|---|---|
|  |  | Dec | Mar | May | Jul | Dec | Mar | May | Jul | Dec | Mar | May | Jul |
| All | BASE | -3.55 | -2.22 | -1.56 | 0.02 | 4.29 | 2.41 | 1.89 | 1.09 | 0.63 | 0.69 | 0.35 | 0.71 |
|  | LC | -3.56 | -2.27 | -1.44 | 0.14 | 4.29 | 2.45 | 1.8 | 1.07 | 0.63 | 0.67 | 0.33 | 0.73 |
|  | PARAM | -3.56 | -1.8 | -0.44 | 1.6 | 4.29 | 2.13 | 1.14 | 1.97 | 0.63 | 0.66 | 0.39 | 0.71 |



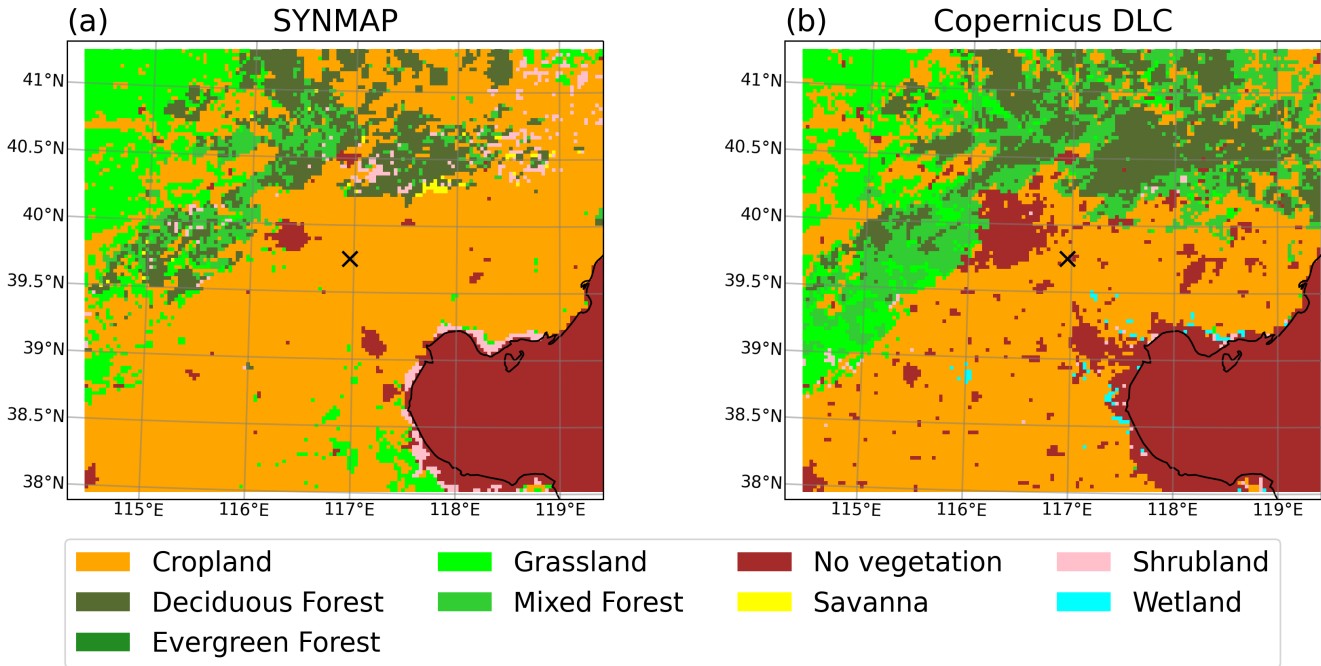

**Figure A3.** Dominant VPRM vegetation category in WRF-GHG domain d03 (3 km) for (a) 1-km SYNMAP and (b) 100-m Copernicus Dynamic Land Cover Collection 3 (epoch 2019). The black cross indicates the location of the Xianghe site.

*Acknowledgements.* We would like to thank all staff at the Xianghe site for operating the FTIR and PICARRO measurements. This work
is supported by the National Key Research and Development Program of China (No. 2023YFB3907500, 2023YFB3907505). Emmanuel
Mahieu is a senior research associate with the F.R.S.-FNRS. The authors acknowledge all providers of observational data and emission
inventories. We thank the IT team at BIRA-IASB for their support on data storage and HPC maintenance. Christophe Gerbig, Roberto
Kretschmer, and Thomas Koch (MPI BGC) are thanked for distributing the VPRM preprocessor code. Finally, we are grateful for fruitful
discussions with Jean-François Müller (BIRA-IASB) and Bernard Heinesch (ULiège).



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
