# Peer review of "A WRF-Chem study of the greenhouse gas column and in situ surface mole fractions observed at Xianghe, China. Part 2: Sensitivity of carbon dioxide (CO2) simulations to critical model parameters"

_EGUsphere, 2025_

## Referee Comment (RC1)

Review comments on "A WRF-Chem study of the greenhouse gas column and in situ surface mole fractions observed at Xianghe, China. Part 2: Sensitivity of carbon dioxide (CO2) simulations to critical model parameters" by Callewaert et al.

The manuscript presents a WRF-GHG simulation framework tailored to Xianghe, evaluating both column-averaged XCO2 and near-surface CO2. The model reproduces the temporal variability of XCO2 with good skill (r≈0.7), despite a persistent background bias, and robustly captures a July 2019 heatwave anomaly. Near the surface, the model performs well in the afternoon ( $r\approx0.75$ , MBE $\approx-1.65$  ppm after bias correction), while nighttime CO2 is consistently overestimated (MBE≈6.5 ppm), yielding an exaggerated diurnal amplitude (~22 ppm observed, overestimated by ~4.6 ppm). Sensitivity experiments show that using elevated anthropogenic emission profiles reduces nighttime overestimation and improves diurnal amplitude agreement, with particularly strong improvements in March and July (e.g., amplitude overestimation reduced from ~22.7 ppm to ~1.7 ppm in March; ~14.2 ppm to ~-6.0 ppm in July). These changes have minimal impact on XCO2. Land cover choice and VPRM parameter adjustments also affect nighttime near-surface CO2 and seasonal behavior, with industry and energy dominating anthropogenic contributions and biogenic fluxes providing a moderate summer sink. The manuscript demonstrates strong diagnostic capability (e.g., tracer partitioning, diurnal PBL analysis), and provides plausible explanations for observed/model discrepancies.

The paper is well structured, clearly written, and the experiments are well designed. The tracer framework and sensitivity analyses are valuable, and the case study is compelling. Most important, the authors reorganize the limitations of the model setup and have sufficient discussion around those limitations. A few important clarifications and additions—particularly regarding vertical sampling, transport evaluation, and the vertical distribution of CO2—would substantially strengthen the interpretation and generality of the conclusions.

Recommend publication after minor revisions. The comments below aim to improve clarity, document assumptions, and bolster the evidence base for key claims.

**General comments:**

- The striking difference in biosphere contributions between XCO2 and in situ CO2 in Figure 5 warrants explicit analysis of the vertical distribution of simulated CO2 and tracer components. Please consider to include some or all the following items in the revision:
  - Vertical profiles and/or cross-sections of total CO2 and tracer-specific contributions (background, anthropogenic sectors, biosphere) during representative periods (e.g., the July event and a typical spring day).
  - PBLH overlays and stability indicators to relate vertical gradients to mixing state.
  - A cross-section similar to Figure A1 focused on the biosphere tracer, and ideally analogous profiles for anthropogenic tracers, to show how vertical gradients translate into different column vs. surface signatures.

These figures will strengthen your explanation of partitioning differences and also provide visual evidence for the roles of PBL mixing and emission release height discussed elsewhere.

- 2. Please clarify the sampling level used to compare model outputs to in situ measurements:
  - Is the comparison at the first model level, interpolated to the instrument height, or sampled as a layer-average?
  - Consider to provide a short sensitivity test on vertical sampling (e.g., first vs. second model level; interpolation to sensor height) to demonstrate robustness. This is conceptually parallel to the sensitivity to emission release levels and may influence nighttime biases in stable conditions.
- 3. Transport is often the dominant source of bias in CO2 simulations. While you diagnose biases from initial conditions and fluxes, a targeted transport evaluation would help establish confidence in the dynamics:
  - Compare simulated meteorology against observations: near-surface wind speed/direction, temperature, humidity, and especially PBL height (ceilometer/radiosonde/reanalysis if available) if they are available
    Demonstrating good transport fidelity will substantiate your focus on flux and

Demonstrating good transport fidelity will substantiate your focus on flux and boundary-condition uncertainties.

**Specific comments:**

- 1. WRF-GHG has 60 vertical levels. Please report how many reside within the typical PBL (<2 km) over Xianghe and the lowest-level thicknesses. Dense resolution in the PBL is crucial for resolving steep CO2 gradients. If possible, discuss whether vertical resolution could influence nighttime bias.
- 2. Why was a two-week spin-up chosen? Please clarify CO2 initial conditions (e.g., from CAMS vs. homogeneous/zero fields). If initialized from non-physical fields, demonstrate that the domain (including vertical extent) reaches a dynamically consistent state post spin-up (e.g., by showing domain-mean CO2 convergence, vertical profile stabilization).
- 3. Just wanted to acknowledge: Good approach to deseasonalize the CO2 time series prior to correlation analysis
- 4. There is a gap in in situ measurements between July and August 2019, coincident with the XCO2 anomaly analyzed in Section 3.3. Please explain the data gap (instrument downtime, QA/QC filtering, etc.) and discuss any implications.
- 5. Table 2 reports nighttime and afternoon metrics. Please also include morning statistics (e.g., 08:00–12:00 LT), as morning transition periods are critical for entrainment and can reveal transport/flux issues.
- 6. Consider combining Table 2 and 3 and using parentheses to show values after bias correction, which will make the presentation more compact and reader-friendly.
- 7. You apply a mean bias correction derived from TCCON sites in 30–50°N. Please justify this choice. Given strong spatial gradients in XCO2, a site-specific CAMS bias at Xianghe may be more appropriate.
- 8. The discrepancy between XCO2 and in situ biosphere contributions is puzzling given that other tracers track similarly across perspectives. Please add a figure showing vertical gradients (profiles or cross-sections) of biosphere and other tracer CO2 to substantiate the explanation. A cross-section anchored on the region highlighted in Figure A1 would be ideal.

- 9. Line 232: Remove the phrase "which is linked to the northern hemisphere's growing season and increased photosynthesis."
- 10. Lines 235–244: Please clarify how XCO2 and its standard deviations were computed for the "before," "during," and "after" periods. Define the exact time windows, averaging procedure (hourly vs. daily means), and whether uncertainty reflects temporal variability, sampling, or retrieval error.
- 11. It would be informative to overlay prevalent wind direction in Figure 6c to connect transport pathways with tracer anomalies.
- 12. Figure A3: This is a useful figure but appears not to be referenced in the main text. Please add a citation and incorporate its interpretation where relevant.
- 13. The authors claimed that the high biosphere signal is attributed to sources in the Gobi Desert/Inner Mongolia, but, in Figure A1, winds appear southerly and Xianghe is near a high-pressure center, consistent with elevated temperatures. Please revisit the source attribution in light of the wind fields and temperature pattern shown.
- 14. Consider adding a vertical gradient plot of total CO2 and relevant tracers to support statements about PBL mixing and vertical distribution.
- 15. Many PBL/surface-layer schemes can produce sharp near-surface gradients under stable conditions. Please check whether this artifact occurs in your simulation and whether it contributes to nighttime bias.
- 16. For the sentence stating improved agreement with elevated emissions "particularly in March and July," please add reasoning or speculation (seasonal stability, emission sector timing, boundary-layer depth) if direct evidence is limited.
- 17. Consider including spatial maps of VPRM fluxes in Figure A1, which would make the biosphere signal interpretation more transparent.
- 18. Line 390: Please confirm whether this refers to Table A3 or Figure A3 and correct accordingly.

Overall, the manuscript is strong and close to publication. Addressing the vertical distribution, transport evaluation, and a few documentation gaps will substantially enhance clarity and confidence in the conclusions.

---

## Author Comment (AC2)

**Author comments on "A WRF-Chem study of the greenhouse gas column and in situ surface mole fractions observed at Xianghe, China. Part 2: Sensitivity of carbon dioxide ($CO_2$) simulations to critical model parameters"**

We wish to thank the reviewers for their careful reading of our manuscript and their constructive comments. We will address each of the comments below. The reviewer comments are written in black, while our author comments are in blue. Modifications to the manuscript are in cyan.

**1  Reviewer 1**

The manuscript demonstrates strong diagnostic capability (e.g., tracer partitioning, diurnal PBL analysis), and provides plausible explanations for observed/model discrepancies.

The paper is well structured, clearly written, and the experiments are well designed. The tracer framework and sensitivity analyses are valuable, and the case study is compelling. Most important, the authors re-organize the limitations of the model setup and have sufficient discussion around those limitations. A few important clarifications and additions—particularly regarding vertical sampling, transport evaluation, and the vertical distribution of $CO_2$—would substantially strengthen the interpretation and generality of the conclusions.

Recommend publication after minor revisions. The comments below aim to improve clarity, document assumptions, and bolster the evidence base for key claims.

Thank you very much for the positive assessment of our work and for the constructive suggestions, which have helped us further strengthen the manuscript.

**General Comments:**
The striking difference in biosphere contributions between $XCO_2$ and in situ $CO_2$ in Figure 5 warrants explicit analysis of the vertical distribution of simulated $CO_2$ and tracer components. Please consider to include some or all the following items in the revision :

- Vertical profiles and/or cross-sections of total $CO_2$ and tracer-specific contributions (background, anthropogenic sectors, biosphere) during representative periods (e.g., the July event and a typical spring day).

- PBLH overlays and stability indicators to relate vertical gradients to mixing state

- A cross-section similar to Figure A1 focused on the biosphere tracer, and ideally analogous profiles for anthropogenic tracers, to show how vertical gradients translate into different column vs. surface signatures.

These figures will strengthen your explanation of partitioning differences and also provide visual evidence for the roles of PBL mixing and emission release height discussed elsewhere.

We agree that the contrasting biospheric contributions to $XCO_2$ and in situ $CO_2$ deserve additional attention. As detailed in the related comment below, we have now added a dedicated section that explicitly discusses this aspect and includes a new figure showing vertical profiles of biogenic $CO_2$ throughout the year, as well as annual mean profiles for the other tracers. We believe this addition provides the necessary visual evidence

to support our interpretation of the partitioning differences.

Regarding the influence of PBL mixing and emission release height, we have also expanded the Discussion with a dedicated subsection that examines their roles in simulating near-surface $CO_2$ (see also related comments below). We hope these revisions address the reviewer's concerns and strengthen the manuscript accordingly.

Please clarify the sampling level used to compare model outputs to in situ measurements:

- Is the comparison at the first model level, interpolated to the instrument height, or sampled as a layer-average?

- Consider to provide a short sensitivity test on vertical sampling (e.g., first vs. second model level; interpolation to sensor height) to demonstrate robustness. This is conceptually parallel to the sensitivity to emission release levels and may influence nighttime biases in stable conditions.

The observations are taken at 60 m above ground level. To enable a relevant comparison with the model, we interpolate the model profiles to 60 m above ground level as well. This was mentioned in Part 1 (Callewaert et al., 2025) and line 111 of the preprint. In practice, this interpolated point lies between the two lowest layer mid-points, since their respective layer thickness is about 50 m and 64 m, see Fig. A. Indeed, the sampling level has a significant impact on the results, see Fig. B, especially at night when there is a strong near-surface gradient (Table 1).

As a result of a comment by the other reviewer, we have restructured our Results section (see below). In the new version, we propose to add a subsection discussing the remaining uncertainties in accurately simulating near-surface $CO_2$ that are not covered by the sensitivity experiments. This contains the choice of PBL scheme within WRF, but also the vertical model resolution (see also later comment). Specifically, we would add the following sentences related to sampling height:

Another contributing factor is the vertical representation of the atmosphere near the surface. Since the observations are collected at 60 m above ground, the simulated $CO_2$ fields were interpolated to this height. However, in our configuration, the difference in simulated nighttime $CO_2$ between the two lowest model layers (about 50 and 64 m thick) can reach 20 ppm. This implies that combining an interpolation with too coarse a resolution may introduce errors of several ppm. Increasing vertical resolution within the PBL would help to reduce these artifacts.

Table 1: Statistical metrics for nighttime (22-4 LT) $CO_2$ at Xianghe, for different sampling heights. Bias corrected values are given between brackets.

|  | level 0 | 60 mgl | level 1 |
|---|---|---|---|
| MBE | 19.41 (20.10) | 7.18 (7.86) | -0.06 (0.62) |
| RMSE | 34.98 (35.42) | 23.77 (24.04) | 22.34 (22.45) |
| CORR | 0.54 (0.54) | 0.60 (0.60) | 0.59 (0.59) |

Transport is often the dominant source of bias in $CO_2$ simulations. While you diagnose biases from initial conditions and fluxes, a targeted transport evaluation would help establish confidence in the dynamics:

- Compare simulated meteorology against observations: near-surface wind speed/direction, temperature, humidity, and especially PBL height (ceilometer/radiosonde/reanalysis if available) if they are available

Demonstrating good transport fidelity will substantiate your focus on flux and boundary-condition uncertainties.

We fully agree that evaluating the simulated meteorology against observational data is important for building confidence in the model results. However, due to the limited availability of publicly accessible meteorological datasets for our study domain, a comprehensive evaluation within the scope of this manuscript was not feasible. We partly addressed this aspect in our companion paper (Callewaert et al., 2025, p. 9524), by comparing the model output with the Global Hourly – Integrated Surface Database (ISD) provided by the National Centers for Environmental Information (NCEI). The results indicated reasonable agreement between the simulations and observations, considering the reported precision. While the exact comparison results are not

[Figure]

Figure A: Illustration of vertical structure near the surface within WRF-GHG at Xianghe, indicating layer thickness (left) and altitude of model levels (solid lines, right) and mid-layer points (dashed lines, crosses, right). Remark that the vertical structure is not fixed over time since WRF implements a hybrid vertical coordinate (terrain-following near the surface, isobaric at 50 hPa). Therefore the shown values are the mean altitudes, averaged over the complete time series. The surface altitude within WRF-GHG at Xianghe is 16.6 m above see level. The green triangle indicates the observation altitude: 60 m above ground level.

included in the manuscript, they are documented in the peer review author comments associated with the companion paper. Given this prior evaluation, we consider it redundant to replicate the analysis here.

**Specific Comments:**
WRF-GHG has 60 vertical levels. Please report how many reside within the typical PBL ($<2$ km) over Xianghe and the lowest-level thicknesses. Dense resolution in the PBL is crucial for resolving steep $CO_2$ gradients. If possible, discuss whether vertical resolution could influence nighttime bias.
There are 11 layers below 2 km, where the lowest level has a thickness of about 50 m. We will add the following to the model description section:'.., and 60 vertical levels extending from the surface up to 50 hPa. There are 11 layers in the lowest 2 km, with a layer thickness ranging from about 50 m near the surface, to 400 m above 2 km'.
In our configuration, the lowest model layer is approximately 50 m thick, and the second layer is about 64 m thick, as also mentioned above and shown in Figure A. To compare with observational data taken at 60 m above ground level, we interpolate the model profile (which is defined at layer mid-points) to this height, which essentially lies in the lower half of the second model layer. Nighttime $CO_2$ profiles typically show strong vertical gradients near the surface due to the stable nocturnal boundary layer. The vertical resolution of our model may be insufficient to fully capture these gradients. A too coarse resolution can lead to artificial vertical mixing, which tends to smooth out sharp near-surface gradients. However, the impact of vertical resolution also depends on the sampling height relative to the layer structure. In the case of a steep negative gradient, layer-averaged values (represented by mid-layer concentrations) may underestimate $CO_2$ near the bottom of the layer and overestimate it near the top. Since we interpolate between these mid-layer values, it is possible that this approach leads to an overestimation of nighttime $CO_2$ at the sampling height. However, the exact impact is difficult to assess without conducting additional sensitivity experiments using higher

[Figure]

Figure B: Hourly median and interquartile range of the and observed (black dashed) and simulated (solid colors) surface $CO_2$ mole fraction at Xianghe, at different sampling heights.

vertical resolution.
As replied to the related general comment above, we have added a small discussion on the impact that vertical model resolution, and sampling height may have on the nighttime bias (see above).

Why was a two-week spin-up chosen? Please clarify $CO_2$ initial conditions (e.g., from CAMS vs. homogeneous/zero fields). If initialized from non-physical fields, demonstrate that the domain (including vertical extent) reaches a dynamically consistent state post spin-up (e.g., by showing domain-mean $CO_2$ convergence, vertical profile stabilization).
The spin-up phase is implemented to mitigate numerical artifacts introduced during model initialization and to ensure mixing of surface-atmosphere fluxes across tracer fields. This was briefly mentioned in the accompanying paper: "This conservative spin-up period is implemented to ensure thorough mixing of the tracers within the domain." (Callewaert et al., 2025). To address the first concern, numerical instability, a few hours of simulation time typically suffice. For the second goal, achieving adequate mixing, the duration is loosely informed by the time it takes for a standard weather system to traverse the model domain. Once that interval has passed, boundary conditions and surface fluxes are expected to be reasonably distributed throughout the simulation space. While a few days would likely be adequate, a two-week spin-up is adopted as a cautious buffer to ensure robustness.
As mentioned in section 2.2 ('WRF-GHG model simulations', line 102-103 of the preprint), the CAMS global reanalysis EGG4 dataset was chosen as initial conditions. Its aim is to represent a realistic distribution of tracers in the atmosphere at the start of the model simulation. The evolution of this information over time is contained in a background tracer. The other tracer fields, representing different flux categories, are initialized at zero or at a fixed value of 50 ppm for those tracers that represent uptake (biosphere and ocean flux). This offset is subtracted in post-processing.
Figure C illustrates the temporal evolution of simulated $CO_2$, averaged across the entire model domain d01 (both horizontally and vertically). Note that data from the first day of the spin-up period (15 August 2018) is unfortunately unavailable as the data was accidentally removed from our servers.
In the first 4–5 days, the background and biogenic $CO_2$ tracer steadily drop, while at the same time, the anthropogenic tracer shows a small but consistent increase. This behavior reflects the model adjusting from its initial conditions: the background tracer drops as the high near-surface $CO_2$ from the initialization gets flushed out, while the anthropogenic tracer builds up gradually as it mixes and accumulates across the do-

[Figure]

Figure C: Temporal evolution of $CO_2$ tracer contributions averaged over d01: background $CO_2$ in blue (left axis), anthropogenic $CO_2$ in red (right axis) and biogenic $CO_2$ in green (right axis). Both y-axis have the same scale.

main. After these first few days, the system settles into a more typical pattern, with expected temporal variability. The biogenic tracer, in particular, shows noticeable fluctuations, which align with strong biospheric activity during August.

Just wanted to acknowledge: Good approach to deseasonalize the $CO_2$ time series prior to correlation analysis.
Thank you.

There is a gap in in situ measurements between July and August 2019, coincident with the $XCO_2$ anomaly analyzed in Section 3.3. Please explain the data gap (instrument downtime, QA/QC filtering, etc.) and discuss any implications.
The in situ $CO_2$ and $XCO_2$ time series are recorded by two entirely independent measurement systems, meaning that a malfunction in one does not compromise the stability of the other—except in cases where atmospheric conditions influence both. The gap in the in situ $CO_2$ data between July 29 and August 21, 2019, is attributed to instrument failure, as mentioned in section 3.1 of Yang et al., 2021. We will add the following: Note that there is a gap in the in situ $CO_2$ time series during this period due to instrument malfunctions.

Table 2 reports nighttime and afternoon metrics. Please also include morning statistics (e.g., 08:00–12:00 LT), as morning transition periods are critical for entrainment and can reveal transport/flux issues.
Done. Table 2 has also been merged with Table 3 (see next comment), and hourly ranges for night and afternoon were adjusted based on comments of the second reviewer (to 22h - 4h LT and 12h - 15h LT, respectively), see updated Table 2.

Table 2: Updated table 2: including statistics for morning transition, updated values for modified afternoon and nighttime periods and merged with table 3 by adding the bias-corrected values between brackets.

|  | insitu $CO_2$ | | | $XCO_2$ |
|---|---|---|---|---|
|  | afternoon | night | morning transition |  |
| MBE | -3.12 (-2.44) | 7.18 (7.86) | -0.73 (-0.04) | -1.43 (-0.86) |
| RMSE | 15.35 (15.27) | 23.77 (24.04) | 22.04 (22.15) | 2.45 (1.80) |
| CORR | 0.75 (0.76) | 0.60 (0.60) | 0.69 (0.69) | 0.70 (0.70) |

Consider combining Table 2 and 3 and using parentheses to show values after bias correction, which will make the presentation more compact and reader-friendly.
This is a good suggestion, we have altered the tables accordingly.

You apply a mean bias correction derived from TCCON sites in 30–50°N. Please justify this choice. Given strong spatial gradients in $XCO_2$, a site-specific CAMS bias at Xianghe may be more appropriate.
We use the CAMS dataset as initial and lateral boundary conditions in WRF-GHG, capturing external influences beyond the d01 domain. Internal fluxes are represented by WRF-GHG tracers. Our comparison of monthly mean TCCON and CAMS $XCO_2$ values revealed consistent seasonal biases across multiple sites, with minor site-specific variations, indicating a systematic bias in CAMS. To estimate a representative bias for latitudes 30–50°N, we averaged across all sites in this range. Using Xianghe alone would be misleading, as its bias may reflect local flux inaccuracies specific to CAMS, which are not part of our system.

The discrepancy between $XCO_2$ and in situ biosphere contributions is puzzling given that other tracers track similarly across perspectives. Please add a figure showing vertical gradients (profiles or cross-sections) of biosphere and other tracer $CO_2$ to substantiate the explanation. A cross-section anchored on the region highlighted in Figure A1 would be ideal.
This is indeed an interesting aspect of our results. To support our claim that the observed differences between $XCO_2$ and in situ biosphere contributions stem from varying measurement sensitivities, we propose including a figure showing the mean vertical profiles of biogenic $CO_2$ at Xianghe (see Fig. Da). The figure reveals three key insights:

- The biogenic tracer contributions are predominantly located below 4 km altitude.

- Between May and September, the $CO_2$ signal is generally negative throughout the column, except near the surface—indicating widespread vegetation uptake in the surrounding region.

- Near-surface values (below 200 m) are, on average, positive in all months except August.

This figure complements Fig. 5 in the preprint, which presents monthly mean tracer contributions by flux category. There, we observe a negative biosphere contribution in August for in situ $CO_2$, while $XCO_2$ consistently reflects a biospheric sink from May through September. Together, these profiles suggest that the discrepancy arises from differences in signal near the surface versus higher altitudes, highlighting the role of measurement sensitivity.
To further illustrate why such a discrepancy is not observed for the other tracers, we include a similar figure showing the mean vertical profiles of all tracers averaged over the full simulation period (see Fig. Db). This figure shows that, like the biogenic tracer, the main contributions from other source sectors are also concentrated below 4 km. However, unlike the biogenic tracer, these contributions are uniformly positive throughout the vertical column. In particular, the energy, industry, and residential sectors exhibit strong signals near the surface that decay exponentially with altitude. Because these tracers do not exhibit a sign change with height, both simulated in situ $CO_2$ and $XCO_2$ capture relatively similar contributions from these sources.
We propose to add a separate Discussion section in the revised manuscript, which would include a subsection discussing this result, titled "Sector contributions: differences between in situ $CO_2$ and $XCO_2$":
In Section 3.2 we compared relative tracer contributions in the WRF-GHG simulations for near-surface $CO_2$ and column-averaged $XCO_2$, and found a notable difference in the biogenic contribution between the two. To analyze this difference, Fig. D presents mean vertical profiles of the simulated $CO_2$ tracers at Xianghe. The profiles show that the dominant tracer signals in WRF-GHG are limited to the lowest 4 km of the column. Panel (a) reveals a large monthly variability in the vertical distribution of the biogenic tracer. From May through September the biogenic signal at Xianghe is generally negative through much of the column but positive in the lowest levels. Near-surface values (below 200 m) are, on average, positive in all months except August. This pattern is consistent with Fig. 5, which shows a negative biosphere contribution in August for in situ $CO_2$ while $XCO_2$ indicates a biospheric sink across May–September. These vertical profiles indicate that the difference can be linked to two factors: the different sensitivities of the measurement techniques and Xianghe's location relative to strong land sinks. In situ observations are typically more sensitive to local fluxes

[Figure]

Figure D: Vertical distribution of simulated $CO_2$ tracers in WRF-GHG at Xianghe up to 12 km altitude (a) at monthly scale for the biogenic tracer and (b) annual scale for all tracers.

(i.e. from urban areas and cropland), which are a net source for most months (except August) as calculated by VPRM (see Fig. 6). In contrast, column measurements ($XCO_2$) integrate the entire atmosphere and are sensitive to fluxes on a larger scale: in this case the forested mountains roughly 50 km north and 90 km east of Xianghe (see Fig. 6), producing a sink over summer.

Panel (b) of Fig. D shows mean profiles of all tracers averaged over the full simulation period. Unlike the biogenic tracer, the industry, residential, and energy tracers are positive at all heights and exhibit a strong near-surface maximum that decays exponentially with altitude. Because these anthropogenic tracers do not change sign with height, their relative contributions are similar for both in situ $CO_2$ and $XCO_2$.

Line 232: Remove the phrase "which is linked to the northern hemisphere's growing season and increased photosynthesis."
Done.

Lines 235–244: Please clarify how $XCO_2$ and its standard deviations were computed for the "before," "during," and "after" periods. Define the exact time windows, averaging procedure (hourly vs. daily means), and whether uncertainty reflects temporal variability, sampling, or retrieval error.

As stated on line 233, this section focuses on the WRF-GHG simulated values between 7 July 2019 and 30 August 2019 to explore the increased $XCO_2$ between 20 and 29 July. Therefore the time windows are as follows: before (7-19 July), during (20-29 July) and after (30 July - 30 August). In the preprint, we calculated these values based on the hourly model simulations that are paired with TCCON measurements, corresponding with the dots in panel 7b. However, as we aim to represent the temporal variability within each period, we agree that it might be more accurate to calculate these values on all hourly model simulations, not only those paired with observations. This leads to slightly higher $CO_2$ mole fractions before and after the event, and larger variability. However the overall pattern remains, where $XCO_2$ first increases a few ppm and then drops. We will modify this in the manuscript as follows: As shown in Fig. 7a, the total simulated $XCO_2$ increases from $406.30 \pm 1.97$ ppm before the summer spike (7–19 July) to $408.23 \pm 1.67$ ppm during the spike (20–29 July), then decreases to $405.01 \pm 1.63$ ppm afterward (30 July–30 August). These values represent the mean and standard deviation of all hourly WRF-GHG simulated $XCO_2$ values during each period.

For consistency, we updated the mean values further in the text: ... Figure 7a shows that the background

[Figure]

Figure E: Figure 6 from preprint with wind barbs overlayed on panel c. These barbs represent the mean wind direction and speed at 800 hPa over the whole simulation period.

$XCO_2$ remains relatively constant in July (406.65 $\pm$ 0.93 ppm), and decreases to 405.63 $\pm$ 1.19 ppm in August. and: ... Thus, the increase in $XCO_2$ between 20-29 July is mainly linked to a weaker biogenic sink (-0.86 $\pm$ 1.04 ppm) compared to the periods before (-3.56 $\pm$ 1.44 ppm) and after (-2.19 $\pm$ 1.39 ppm).

It would be informative to overlay prevalent wind direction in Figure 6c to connect transport pathways with tracer anomalies.
We appreciate the suggestion and agree that visualizing the dominant wind directions could, in principle, help relate transport pathways to tracer anomalies. However, the prevailing wind direction at Xianghe varies substantially throughout the year. When averaged over the full annual period, this variability collapses into a predominantly easterly mean flow (see Fig. E), which provides little meaningful insight into actual transport pathways. We therefore believe it is preferable to keep Fig. 6c in its current form, without wind barbs.

Figure A3: This is a useful figure but appears not to be referenced in the main text. Please add a citation and incorporate its interpretation where relevant.
This figure was referenced in the section on 'Biogenic flux and land cover sensitivity': ...comparison of the two land cover datasets over the innermost WRF-GHG domain (d03) is shown in Figure A3.

The authors claimed that the high biosphere signal is attributed to sources in the Gobi Desert/Inner Mongolia, but, in Figure A1, winds appear southerly and Xianghe is near a high-pressure center, consistent with elevated temperatures. Please revisit the source attribution in light of the wind fields and temperature pattern shown.
We appreciate the reviewer's careful reading and feedback; however, after re-examining the multi-day maps we maintain our original attribution that advection of a warm, low-biogenic-uptake airmass from the west/northwest, followed by locally enhanced respiration, explains the elevated $XCO_2$ at Xianghe between 20 and 29 July 2019.
To clarify this we added 800 hPa geopotential contours to the potential-temperature maps and extended the analysis period (Figs. G and H). The synoptic evolution is as follows: before the event (19 July 2019) Xianghe is situated between a northern low and Tropical Storm Danas to the southeast, producing a shear wind flow. As the northern low intensified and Danas moved northeast, a northwesterly flow developed, advecting a high-potential-temperature airmass (with weak biogenic uptake) towards Xianghe; the onset of

that advection coincides with the observed jump in the biogenic $XCO_2$ tracer on 20 July. Winds remain predominantly westerly/northwesterly over the core of the event, transporting the warm, low-uptake air eastward, while local temperatures rise and ecosystem respiration increases. A brief southerly episode on 22 July is indeed shown in the maps and corresponds to a concurrent spike in anthropogenic tracer signals and a temporary change in the biosphere fluxes (Fig. 7b); this short phase does however not change the overall advection-driven attribution. Later in the event (from 26 July onwards), the 800 hPa winds turn more southwesterly; however, the warm airmass has already arrived and is being spread northeastward, maintaining locally elevated temperatures. Overall, the combined evidence from 800 hPa winds, potential temperature, the biogenic $XCO_2$ field, and NEE in Figs. G and H supports our conclusion that the elevated $XCO_2$ results from two related but separable processes: an immediate increase due to horizontal advection of a warm, low-uptake airmass, followed by enhanced local respiration.

We propose to add the two modified figures of the synoptic maps to the manuscript and modify the section as follows (note that we additionally moved this subsection to a new Discussion section, as a result of restructuring the manuscript):

...Synoptic maps (Fig. G and Fig. H) show that at the onset of the event, on 20 July 2019, a warm air mass arrived from the northwest. This air mass, originating from the Gobi Desert and grasslands in Inner Mongolia, both areas that are characterized by sparse vegetation and elevated temperatures, carried a lack of biogenic signal and coincides with the jump in the biogenic $XCO_2$ tracer. Additionally, the mean NEE around Xianghe, as calculated by VPRM, is slightly higher between 20 and 29 July (average of -5 941 mol $km^{-2}$ $h^{-1}$ over domain d03) compared to the periods before and after (respectively -9 153 and -12 785 mol $km^{-2}$ $h^{-1}$). ... Therefore, we conclude that the spike was caused by an atmospheric circulation anomaly resulting in the advection of a warm air mass with high biogenic $CO_2$ levels, followed by locally reduced photosynthesis and increased respiration due to the resulting hot temperatures.

Consider adding a vertical gradient plot of total $CO_2$ and relevant tracers to support statements about PBL mixing and vertical distribution.

Unfortunately, the intent of this comment was not entirely clear to us, but we have addressed it to the best of our understanding by adding vertical profiles of biogenic $CO_2$ as well as annual mean profiles of the other tracers (see response above). These plots provide a direct illustration of the vertical distribution and we hope that this addition meets the reviewer's expectations.

Many PBL/surface-layer schemes can produce sharp near-surface gradients under stable conditions. Please check whether this artifact occurs in your simulation and whether it contributes to nighttime bias.

The model indeed produces a pronounced near-surface gradient under stable nighttime conditions (see Fig. F), with a mean decrease of about 20 ppm between the lowest two model layers. Some degree of stratification is expected due to $CO_2$ accumulation within the nocturnal boundary layer. However, without dedicated vertical profiling observations (e.g., ceilometer measurements), which are not available for Xianghe, it is difficult to determine whether the magnitude of this gradient is realistic or an artifact of the PBL/surface-layer scheme. Such an artifact could indeed contribute to the nighttime positive bias, but we cannot robustly quantify this effect with the existing data. In response to this and other comments, we have expanded the Discussion section to more thoroughly address the role of near-surface vertical resolution, sampling height, and PBL representation on the simulated $CO_2$ values.

For the sentence stating improved agreement with elevated emissions "particularly in March and July," please add reasoning or speculation (seasonal stability, emission sector timing, boundary-layer depth) if direct evidence is limited.

The improved agreement in March and July refers specifically to the amplitude of the simulated diurnal cycle. Other statistical metrics show a more mixed picture, as noted earlier in the section. Our results indicate that elevated emissions primarily affect nighttime $CO_2$ concentrations, when the boundary layer is shallow, so their impact is less sensitive to seasonal variability than the daytime signal. In that sense, the key question becomes why similar improvements are not seen in May and December. For May, this is explicitly discussed in the manuscript: the short experimental period combined with high model–observation variability limits the detectability of the sensitivity effect. For December, the reason remains unclear; as outlined in the new Discussion subsection, this may point to remaining uncertainties in the model setup, including whether the

[Figure]

Figure F: Mean vertical $CO_2$ profile at Xianghe between (22-4 LT), on logarithmic y-scale. The dots represent mid-layer values, while the cyan line indicates the height at which the model is interpolated to compare with the observations.

prescribed vertical emission profiles are appropriate for China and consistent with the spatial and sectoral characteristics of the underlying inventory. Finally, in the revised manuscript we now emphasize that other unresolved factors, such as PBL dynamics and near-surface vertical resolution, could also influence the remaining discrepancies.

Consider including spatial maps of VPRM fluxes in Figure A1, which would make the biosphere signal interpretation more transparent.
Done, see Fig. G. As explained above, we have additionally added 800 hPa geopotential contour lines and extended the maps to cover the complete event, see Fig. H

Line 390: Please confirm whether this refers to Table A3 or Figure A3 and correct accordingly.
This refers to Table A3, but has been removed when restructuring the manuscript. More specifically, we have renamed the sensitivity experiments as a reply to a comment of the other reviewer, whereby the results of the LC and PARAM simulations using SFC emissions are omitted from the manuscript.

Overall, the manuscript is strong and close to publication. Addressing the vertical distribution, transport evaluation, and a few documentation gaps will substantially enhance clarity and confidence .

**2 Reviewer 2**

The manuscript presents an overview of the performance of the WRF-GHG against $CO_2$ observations collected for one year near Xianghe, China (within 100 km from Beijing). The authors evaluate the basic model performance and present a series of sensitivity experiments in order to assess the impact of critical configuration settings. The paper is well-written and mostly well-structured, and of high editorial quality.

The authors find that signals emitted from anthropogenic sources (primarily from industry and energy sectors, but with contributions from residential heating & transportation) dominate the local signal variability

in both column and in situ model realizations of the atmosphere. The authors use the framework developed in the tandem paper and presented for CH4 there. Major additions here are the vertical emission structure (important for anthropogenic sources of $CO_2$), and VPRM biosphere model. The evaluation of their performance is interesting. Seasonal biospheric signals (modelled using online VPRM module) were found to be playing a seasonal role, but the sensitivity experiments have shown a critical necessity for update of the VPRM model parameters, with the default settings not providing accurate predictions over China. This is also not surprising considering the incomplexity of the model.

I think the study is overall good report on the simulation results, but my primary concern is that it might be of relatively low interest. The findings do not go much beyond previously published papers (those given in the introduction, but also the companion paper) neither in terms of methods, data used, nor the model techniques. The main innovation as compared to the companion paper is a new gas investigated over East China. The authors conclude that the study "demonstrates the value of the WRF-GHG model framework to interpret both temporal and sectoral variations in surface and column $CO_2$ observations", and further states that model accuracy is strongly dependent on appropriate configurations choices" – but this is known, and I personally feel that the contents might not warrant a full research paper in ACP.

I recommend for the study to be published only after major revision. This should include expansion of the paper material, especially in terms of more robust sources of error evaluation. The impact of the urban areas on the $CO_2$ fluxes also deserves more attention, as we are analyzing an area in the immediate vicinity of one of the largest cities in the world – Beijing. Application of an inversion framework, even a simple form, should be possible with the results available here, and could strengthen the scientific content substantially (see, for example "2.1. Bayesian Synthesis Inversion" in Peylin et al., 2002), allowing for the quantitative source error estimation.

We thank the reviewer for the thoughtful evaluation of our manuscript and for the constructive suggestions. However, we have a different perspective on some parts of the general comment and would like to outline more clearly the contributions and scope of our study.

Regarding the comment that the work may be of relatively low interest, we would like to note that the other reviewer expressed a different view, highlighting the relevance of the tracer-based framework and the value of the combined surface–column analysis. We therefore believe the study addresses topics of continued interest to the ACP community.

The reviewer also mentions that the findings do not substantially go beyond previous work, including the companion $CH_4$ paper. As Part 2 of a paired study, the modelling framework is indeed shared. However, the present paper introduces several new elements. To our knowledge, this is the first time WRF-GHG has been applied to jointly interpret both $XCO_2$ and in situ $CO_2$ at a single site. Moreover, in the revised manuscript we added a dedicated subsection discussing the differing biogenic contributions to $XCO_2$ versus surface $CO_2$. This comparison, linking measurement sensitivity to horizontal and vertical gradients, provides insight that, to our knowledge, has not been shown in previous WRF-GHG studies. The current $CO_2$ paper also includes additional analyses not present in the $CH_4$ paper, such as a synoptic case study and sensitivity experiments targeted at simulating near-surface $CO_2$. Concerning the suggestion to expand the manuscript, we have substantially strengthened it in response to the reviewers' comments, adding new figures, analyses, and explanations. Given the already considerable length of the paper (31-page preprint), we believe the current material is sufficiently comprehensive while maintaining focus.

On the point that the impact of urban areas on $CO_2$ fluxes deserves more attention, we understand from later comments that this refers mainly to urban biogenic $CO_2$ exchange. This component is indeed not explicitly included in our model setup. However, the sensitivity experiment using an alternative, higher-resolution land-cover map (containing more detailed urban green areas) already partially addresses this aspect, and we discuss this in the corresponding subsection. Lastly, the reviewer suggests implementing an inversion framework for source error quantification. While highly valuable in general, we consider this outside the scope of the present study. Incorporating even a simple inversion would shift the focus substantially and exceed the intended scope of this paper.

Overall, we believe the revised manuscript has improved considerably thanks to the reviewers' input, and we hope that the added analyses and clarifications address the main concerns while keeping the paper focused and coherent.

**Specific major comments**

Structure: section 3 is titled "Results", but it contains a subsection "3.5.3. Discussion". I assume, based on the text, that this is meant to be Section 4 "Discussion", and I've put remarks assuming that this was the targeted structure. If this is not meant this way, Sec. 3 should be titled "Results and Discussion" – in which case I ask to adjust responses to my "move to Discussion comments" accordingly.

The subsection previously titled "3.5.3 Discussion" focused specifically on the limitations of the sensitivity experiments and was therefore not intended to serve as the general Discussion section of the manuscript. However, we agree that the current layout may be confusing and have revised the structure accordingly. The manuscript now contains separate Results and Discussion sections. Some subsections have been expanded and reorganized based on the reviewer's comments. The revised Discussion section now includes:

- A section discussing the difference in relative monthly biogenic tracer contributions between in situ $CO_2$ and $XCO_2$, incorporating a few sentences from the original Sect. 3.2, a new figure, and additional discussion as suggested by the other reviewer's comment.

- The subsection on the July event, which has been moved from Results as it fits better within the interpretative discussion.

- A new section discussing both the results and limitations of the sensitivity experiments, integrating content from the original Sects. 3.4 and 3.5. This part also includes a new subsection addressing remaining sources of uncertainty not covered by the experiments, such as the influence of PBL parameterization and vertical resolution.

We believe this revised structure improves clarity and provides a more coherent and comprehensive discussion of key findings and model limitations. A full overview of the new Results and Discussion section is the following:

3. Results

    3.1. Evaluation of one-year simulation

        i. Timeseries and statistical comparison

        ii. Correction of background bias

    3.2. Sector contributions to observed mole fractions

    3.3. Diurnal cycle analysis of in situ data

    3.4. Sensitivity experiments

        i. Emission height sensitivity

        ii. Biogenic flux and land cover sensitivity

4. Discussion

    4.1. Sector contributions: differences between in situ $CO_2$ and $XCO_2$

    4.2. July $XCO_2$ anomaly case study

    4.3. Sensitivity of near-surface $CO_2$ simulations to model configuration choices

        i. Emission height

        ii. Land cover representation

        iii. VPRM parameterization

        iv. Remaining sources of uncertainty

L120-121, tables 1, 6 & 8: (major) I suggest to restructure the naming of the sensitivity experiments. While reading the Results and Discussion sections, I kept being confused by the BASE simulation occurring in SFC and PROF flavour. As it is right now, the clarity would benefit if you had BASE, SFC, LC, PARAM as four simulations for sensitivity experiments. Tables 6 & 8 could then be merged, as the same simulation is referred to as PROF and BASE in these, with same numbers given.

Table 3: Updated table 1, overview of the model configuration for the sensitivity experiments.

| Experiment name | Emission height | Land cover map | VPRM parameters |
|---|---|---|---|
| BASE | SFC | SYNMAP | Li table |
| PROF | PROF | SYNMAP | Li table |
| LC | PROF | Copernicus LC | Li table |
| PARAM | PROF | Copernicus LC | Glauch table |

Table 4: Proposed merged table 6-8 with updated naming of experiments and periods: overview of statistical metrics for the different sensitivity experiments BASE, PROF, LC and PARAM with respect to the observations, per simulation period. Values for MBE and RMSE are given in ppm.

| | | MBE | | | | RMSE | | | | CORR | | | |
|---|---|---|---|---|---|---|---|---|---|---|---|---|---|
| | | Dec | Mar | May | Jul | Dec | Mar | May | Jul | Dec | Mar | May | Jul |
| | BASE | 10.04 | 10.80 | 0.69 | 1.99 | 31.90 | 22.41 | 17.22 | 19.22 | 0.55 | 0.47 | 0.61 | 0.64 |
| All | PROF | -3.91 | 1.38 | -6.40 | -8.27 | 23.51 | 11.47 | 16.08 | 16.82 | 0.53 | 0.49 | 0.62 | 0.67 |
| | LC | -3.92 | -0.20 | -8.04 | -9.72 | 23.52 | 10.73 | 17.20 | 17.30 | 0.53 | 0.48 | 0.58 | 0.69 |
| | PARAM | -1.82 | 4.61 | 0.34 | 0.82 | 23.60 | 13.09 | 13.79 | 13.64 | 0.55 | 0.50 | 0.68 | 0.73 |
| | BASE | -3.96 | -0.51 | -2.67 | -6.70 | 21.59 | 10.49 | 12.22 | 18.76 | 0.58 | 0.63 | 0.12 | -0.11 |
| Afternoon | PROF | -9.28 | -3.09 | -4.53 | -10.17 | 24.42 | 8.88 | 11.92 | 17.11 | 0.47 | 0.66 | 0.11 | -0.03 |
| | LC | -9.29 | -3.80 | -4.54 | -8.77 | 24.43 | 8.85 | 11.90 | 15.91 | 0.47 | 0.66 | 0.13 | 0.03 |
| | PARAM | -8.53 | -0.23 | 2.32 | -0.84 | 23.84 | 9.75 | 11.00 | 13.70 | 0.50 | 0.63 | 0.23 | 0.10 |
| | BASE | 17.00 | 17.75 | 6.38 | 7.64 | 32.42 | 23.81 | 19.21 | 20.71 | 0.69 | 0.55 | 0.54 | 0.35 |
| Night | PROF | 0.08 | 5.37 | -3.15 | -4.86 | 18.57 | 11.14 | 11.69 | 16.12 | 0.72 | 0.49 | 0.70 | 0.43 |
| | LC | 0.06 | 3.17 | -6.56 | -8.77 | 18.58 | 9.51 | 14.06 | 17.26 | 0.72 | 0.49 | 0.61 | 0.46 |
| | PARAM | 3.00 | 8.90 | 2.56 | 2.64 | 19.38 | 13.78 | 11.15 | 14.47 | 0.73 | 0.48 | 0.73 | 0.54 |
| | BASE | -0.79 | 11.49 | -1.44 | -3.51 | 32.79 | 31.13 | 19.83 | 19.34 | 0.46 | 0.39 | 0.39 | 0.52 |
| Morning | PROF | -17.57 | -1.00 | -9.98 | -16.17 | 30.43 | 14.28 | 20.44 | 21.50 | 0.60 | 0.47 | 0.35 | 0.57 |
| | LC | -17.58 | -2.60 | -10.59 | -15.97 | 30.45 | 13.85 | 20.72 | 21.10 | 0.59 | 0.45 | 0.33 | 0.59 |
| | PARAM | -15.66 | 2.21 | -0.66 | -3.87 | 29.27 | 15.18 | 17.09 | 14.44 | 0.58 | 0.50 | 0.46 | 0.62 |

Thank you for pointing out this issue. We propose to rename the experiments as follows (see Table 3): BASE, PROF, LC and PARAM. The merged table 6-8 would then become Table 4.

L161: Division of data into afternoon and nighttime is questionable. The typical daytime that is considered in the $CO_2$ analysis papers 12-15 or 11-15 LT, to assure developed boundary layer throughout the year; Xianghe is only at 40N south, with a distinct seasonal difference in daylight time. In winter, at 18 LT, the stable boundary layer might already be developing, with the surface measurements more fitting into the nighttime category already. It's much worse, however, with what the authors have selected as "nighttime"; in late December the sunrise is at 7.30 am, in summer 5 am is already daytime. As applied, the division mixes very different atmospheric states in terms of PBL dynamics, and the results of analyses are likely to be affected by diurnal cycle of PBLH rather than by, which doesn't help in evaluation of "nighttime" model performance.

Our initial choice of time periods was based on the typical timing of minimum and maximum daily in situ $CO_2$ concentrations as shown in the mean diurnal cycle (Fig. 8b). However, we acknowledge that sunrise and sunset times vary significantly across seasons at Xianghe (40°N), and that our original definitions may have mixed distinct boundary layer regimes, particularly during winter mornings and evenings. In response, we have revised the period definitions to better reflect atmospheric conditions throughout the year. Specifically, we now define the afternoon period as 12–15 LT and the nighttime period as 22–04 LT, which avoids overlap with sunrise and sunset across all seasons. Additionally, we have introduced a morning transition period (08–12 LT) following a suggestion from the other reviewer.

These changes are reflected in the updated Table 2 (merged with the original Table 3), here shown as Table 2. The revised statistics show modest changes: in the afternoon, the MBE and RMSE increase (from –1.65 ppm to –2.44 ppm and from 14.49 ppm to 15.27 ppm, respectively, after bias correction), while the correlation coefficient remains nearly unchanged (0.75 to 0.76). During the night, the MBE increases from 6.51 ppm to

7.86 ppm, RMSE decreases from 25.10 ppm to 24.04 ppm, and the correlation coefficient improves from 0.57 to 0.60.
Additionally, the metrics for the sensitivity experiments have been modified accordingly, see Table 4, as well as the corresponding description of the results.

**Other comments**
L39-L45: Please also provide a reference to Beck et al. here. Note: WRF-GHG is an evolution of WRF-VPRM used by Ahmadov et al. 2007 in the paper you cite. In L44 I suggest to add "(there referred to as WRF-VPRM)" next to Ahmadov et al. citations.
Thank you for the suggestion, we have added both the reference and the clarification.

L75, and also L79-80: Measurement uncertainty over what timescale? A timescale comparable to instantaneous values of the model output should be provided, unless the model output was averaged over hourly time steps.
The model provides hourly output as instantaneous snapshots, not time-averaged values. The observational data used for comparison are also near-instantaneous measurements, closely aligned in time with the model output. The reported uncertainty of the observation system refers to the precision of individual measurements, not to time-averaged values.

L88: How many vertical levels below 3km? What is the thickness of the lowest layer?
See also the similar comment of the other reviewer above. There are 11 layers below 2 km (or 14 below 3 km), where the lowest layer has a thickness of about 50 m. This has been added in the manuscript.

L93: Biomass burning emissions are highly variable in both space and time. How was this assured? What kind of errors are expected?
We include biomass burning emissions from the FINN v2.5 inventory (Wiedinmyer et al., 2023, Fire Inventory from NCAR) as a separate tracer in our model. This inventory provides daily emissions at high spatial resolution (1 km), derived from satellite-detected fire activity and land cover data, allowing us to capture the spatiotemporal variability of biomass burning events. However, we would like to note that the relevance of this question may be limited in our case: as shown in Wiedinmyer et al. (2023), biomass burning activity in our study region is relatively low compared to fire-prone areas such as Southeast Asia, Africa, or South America. Our model results show that biomass burning tracer contributions are negligible compared to the total concentrations. Nonetheless, we acknowledge that uncertainties may remain in the emission estimates. These include potential errors in satellite fire detection, assumptions in emission factors, and uncertainties in burned area estimation. While these factors can introduce regional and temporal biases, their impact on our results is expected to be minimal given the low biomass burning signal in the region.

L99: What does 00:00 UTC correspond to in local time? Why was this time selected?
China Standard Time is 8 hours ahead of UTC, so 00:00 UTC corresponds with 08:00 LT. The model simulations were re-initialized daily at 00 UTC using global meteorological fields, while tracer fields were carried over continuously. This approach is widely used in the modeling community to ensure that the internal meteorology remains closely aligned with reanalysis data, thereby improving confidence in the simulated dynamics and tracer transport. The choice of 00 UTC follows convention in many regional modeling studies (Feng et al., 2016; Park et al., 2018; Pillai et al., 2011). While this timing corresponds to local nighttime in Europe, it does not universally align with nighttime hours elsewhere, for example, 00 UTC is 16:00 local time in California (Feng et al., 2016; Park et al., 2018) and 08:00 in China, our study domain. We acknowledge that this timing may coincide with the morning boundary layer transition in China, which could, in principle, influence near-surface meteorology and tracer behavior. However, we have examined the model output for signs of discontinuities or artifacts at 00 UTC and found no significant anomalies in key meteorological or tracer fields, see Fig. I.

L110: Were results momentary or averaged in time? How does that compare to the observational data?
Both simulated and observed values represent momentary mole fractions. To ensure a meaningful comparison between model and observations, we averaged the observations within ±15 minutes of each model timestamp.

For the column observations, a smoothing procedure was applied on each matched model–observation pair within ±15 minutes of the model timestamp. Subsequently, the results were averaged to obtain a single representative observation and smoothed model value for that time step. This approach preserves the momentary nature of both datasets while reducing noise and temporal mismatch, allowing for a more robust comparison. This aspect is mentioned in our companion paper (Callewaert et al., 2025), however we acknowledge that it would enhance clarity to emphasize the momentary nature of both datasets by including the following sentence in the manuscript: The hourly model output represents instantaneous values, as do the observational measurements. To align the datasets temporally, the observations are averaged around each model output time step, using a ±15-minute window .

L112-113: How is the column above the model top handled in this study? Even if discussed in the Part 1, this should be mentioned here. This is important for comparing against (total column) TCCON measurements, with possibility of significant, and artificial, biases.
We acknowledge that it is important to handle the column above the model top accordingly. As explained in Part 1, this is done by extending the model profile with the TCCON a priori profile before smoothing. The chosen approach can indeed lead to significant biases, however this is expected to be less relevant for $CO_2$ compared to $CH_4$, which is the focus of the current work. We have adjusted the sentence for clarity: ...while for column measurements the model output is smoothed with the FTIR retrieval's a priori profile and averaging kernel, after being extended above the model top with the FTIR a priori profile.

L160: How was the deseasonalization done?
Deseasonalization was performed by fitting the following regression model to the $CO_2$ time series:

$$Y(t) = A_0 + A_1 \cdot t + \sum_{k=1}^{3} \left( A_{2k} \cos(2k\pi t) + A_{2k+1} \sin(2k\pi t) \right), \tag{1}$$

where $t$ represents time. The fitted seasonal and trend components were then subtracted from the original data to obtain the deseasonalized series.

Figure 2 (also concerns Figure 4):

- Consider only keeping the bias corrected version, and move the uncorrected to the supplement. The difference between 2 and 4 is minor and cannot be distinguished with a naked eye.

- I suggest to include all model data in panels a) and b) – it would be interesting to see the model behaviour for the gap periods, especially the July 2019 event, which could shed some light on the temporal extent of the event (since model performs well).

We thank the reviewer for these constructive suggestions. We have updated the figures accordingly, resulting in revised Figs. J and K. The former corresponds to Fig. 4 in the preprint, now displaying all model time steps together with the bias-corrected version of Fig. 2b,d. The latter corresponds to Fig. 2a,c, now also showing all model hours.
We agree that the effect of the bias correction is visually subtle, but in the $XCO_2$ panels it remains distinguishable. We therefore propose retaining this figure in the manuscript but relocating it to the appendix to avoid overloading the figure count.
As a consequence of showing all simulated time steps, the smoothed model data cannot be used in panels (a), and the original $XCO_2$ values are shown instead. This does not affect interpretation, as the smoothing effect on WRF-GHG $XCO_2$ is minimal (mean difference of 0.78 ppm) and not visible at the plotted scale. Panel (c), which displays the model–observation difference, will continue to use the smoothed model series.
As a result of relocating the figures, we modified sections 3.1.1 and 3.1.2 from the preprint accordingly, with the largest impact on Sect. 3.1.1:
Table 2 summarizes the comparison between simulated and observed $CO_2$ mole fractions at Xianghe. Overall, WRF-GHG demonstrates a reasonable accuracy in replicating these measurements: the $XCO_2$ observations are slightly underestimated, with a mean bias error of -1.43 (± 1.99) ppm and a Pearson correlation coefficient of 0.70. Note that the $XCO_2$ time series was de-seasonalized before calculating the correlation coefficient in

order to remove the effect of the seasonal variation. After applying a bias correction to the modeled values, the $XCO_2$ MBE decreases to –0.86 ($\pm$ 1.57) ppm (corrected values shown in parentheses in Table 2), and the RMSE improves from 2.45 ppm to 1.80 ppm, while the correlation remains unchanged. Details of the bias correction are provided in Sect. 3.1.2, and the corresponding time series are shown in Fig. J (and Fig. K). The data near the surface has been divided into afternoon (12:00 - 15:00 LT), nighttime (22:00 - 04:00 LT) and morning (08:00 - 12:00 LT) periods to assess model performance under different boundary layer conditions. Indeed, WRF-GHG shows a smaller bias (-2.44 ppm) during the afternoon, when the lower atmosphere is well-mixed, compared to nighttime (7.86 ppm). Additionally, the MBE differs in sign between the two periods: near-surface $CO_2$ levels tend to be underestimated by the model in the afternoon but overestimated at night. Except for the moderate correlation observed for in situ $CO_2$ during nighttime (0.60), WRF-GHG achieves relatively high correlation coefficients ($\geq$ 0.7) for other $CO_2$ data, indicating satisfactory model performance. Overall, the bias correction has only a minor influence on the comparison with near-surface mole fractions, where the effect on RMSE and correlation coefficients are negligible ($<$ 1.2% and $<$ 0.01, respectively).

Finally, the $XCO_2$ time series in Fig. Ja reveals a notable spike between 20–29 July (highlighted in gray), interrupting the general decline associated with northern hemispheric photosynthetic uptake during the growing season, from May onwards. A dedicated analysis of this July $XCO_2$ event is provided in Sect. 4.2. Note that there is a gap in the in situ $CO_2$ time series during this period due to instrument malfunctions (Yang et al., 2021).

L166: Which correlation coefficient? Pearsons? R? R2? Be precise.
All correlation coefficient values reported in our study refer to the Pearson correlation coefficient. We will add this clarification at the first mention of the term in the manuscript to ensure transparency.

L175-176: "This bias likely originates from a similar error in the background data, inaccuracies..." – conclusion precedes the results. Remove here and move to discussion please.
We agree that the explanation of the origin of the $XCO_2$ bias includes interpretational elements that could also be appropriate for the Discussion section. However, the outcome of this analysis directly leads to the practical definition of the bias correction that is subsequently applied to all model results throughout the manuscript. This correction is not a simple empirical adjustment based on model–data mismatch, which is a common approach, but is instead derived from identifying the underlying cause of the bias, namely the bias in the CAMS fields used as background. For clarity and continuity, we believe it is important for readers to encounter the reasoning and methodology behind the bias correction at the point where the corrected data are first introduced, rather than postponing this explanation to a later section.
Nevertheless, in response to this and other related comments, we have revised and reorganized the manuscript to strengthen the separation between Results and Discussion in the subsequent sections.

L201: I disagree that MBE changes only "slightly". That's 30% of absolute bias, and 0.7 ppm can be significant.
Thank you for pointing this out. We agree that the change in mean bias error is not negligible. To reflect this, we have removed the word "slightly" from the sentence to avoid understating the significance of the change.

L202-203: Negligible – please provide numbers (can be "smaller than XXX") for reproducibility.
We have added quantitative values to our statement: The effect on RMSE and correlation coefficients for in situ $CO_2$ are negligible ($<$ 1.2% and $<$ 0.01, respectively).

Figure 3: is this needed? Consider moving to supplement.
We appreciate the suggestion to move the figure to the supplement. However, we believe it plays an important role in supporting the discussion in that section. In particular, it provides clear visual evidence of the bias in the CAMS reanalysis relative to several TCCON sites, which strengthens the interpretation of our results. For this reason, we consider it valuable to retain the figure in the main text.
As part of our response to other comments, we have already reduced this section by removing the accompanying Table 3 and Figure 4 from the main text—either by merging Table 3 with Table 2 or relocating Figure

4 to the appendix.

L219: "VPRM-computed fluxes suggest..." – this requires some comment. What kind of crops are grown there? Is such a net flux from croplands realistic? What about the urban-area fluxes?
The area surrounding Xianghe is dominated by irrigated croplands under a winter-wheat/summer-maize double-cropping system. Other species such as sorghum, millet, oil-bearing crops, vegetables, and sweet potatoes are also present. It is well known that the VPRM cropland class has the largest uncertainty because it aggregates many crop types with distinct phenological and biogenic $CO_2$ signals. This uncertainty is amplified at Xianghe due to the lack of locally calibrated VPRM parameters (as stated elsewhere), which is mainly related to the absence of a sufficiently dense eddy covariance flux tower network in the region. Therefore, while the simulated cropland fluxes may not perfectly represent reality, they currently remain the best available estimate.
Regarding urban-area fluxes: the land cover dataset used in the main simulations (SYNMAP) provides only a coarse representation of the heterogeneous land cover around Xianghe and likely underestimates the fraction of urban vegetation. Higher-resolution land cover products would capture this heterogeneity more accurately and include the biogenic $CO_2$ fluxes from urban green spaces. Both these elements are discussed in the new subsection regarding sensitivity of near-surface $CO_2$ simulations to model configuration choices.
The sentence in question was intended specifically to explain the difference between the biospheric contribution in the column versus the near-surface. The Xianghe site is surrounded mainly by cropland and urban areas, whereas the closest strong land sinks—forested mountains—are located tens of kilometres away. Because the FTIR column is more sensitive to these distant sinks than the in situ measurements, the simulated biospheric contribution differs accordingly. This spatial contrast in vegetation types is well supported by the land cover maps, even if the precise flux magnitudes contain uncertainty. In the revised manuscript, this explanation has been moved to the discussion section, where we now also include a new figure showing the vertical distribution of biogenic contributions in WRF to help clarify this point.

L235: "... ppm before... after" – please state explicitly which periods were used as "before" and "after"
In line with a suggestion from the other reviewer, we have clarified these periods by including the exact dates in brackets.

L239-240: "enhacement below / above the background" -¿ negative / positive enhancement
We have modified this in the manuscript: It further clearly indicates a negative contribution of the tracers before and after the summer spike to a positive enhancement during the spike period.

L242: "a less strong" -¿ weaker
Modified.

L244 onwards: from here the text turns into Discussion and should be moved to appropriate section of the paper.
As a result of restructuring the revised manuscript, we have moved the entire July event section to the Discussion.

L247: "a warm air mass with relatively high biogenic $CO_2$ levels," this sounds as if there is a lot of biogenic signal from respiration, whereas in fact this is a lack of signal. Needs rewording.
We have modified the part of the sentence to: .. suggest eastward advection of a warm air mass with a lack of biospheric activity, ..

L249 (and elsewhere in the text): "biogenic $CO_2$ flux" – consider referring to it as "NEE" as in the original VPRM paper, or "net biogenic $CO_2$ flux". Especially the latter would improve clarity for the reader.
We have changed most occurrences of biogenic $CO_2$ flux in the main text to net biogenic $CO_2$ flux or NEE.

L261-275: For clarity of text, I suggest to give the absolute values only for observations and discuss the model over/underestimation together with model results presented as differences. Will some restructuring of the paragraph, but would make for easier reading.

We thank the reviewer for this suggestion. We have rewritten lines 262-272 to the following:

The PBL reaches its maximum height around 15:00 local time (LT), coinciding with the lowest surface $CO_2$ mole fractions. Conversely, during the night, radiative cooling leads to the formation of a stable, shallow PBL, trapping $CO_2$ near the surface and causing mole fractions to rise. As the sun rises and the PBL height begins to increase again, the $CO_2$ mole fractions drop, giving rise to a characteristic diurnal cycle.

Indeed, the lowest values are observed between 14:00 and 16:00 LT, with a minimum of 421.32 (415.80 – 431.76) ppm at 16:00. In the early morning, the observed $CO_2$ mole fractions show a distinct peak at 7:00, reaching 443.42 (428.00–459.32) ppm. WRF-GHG successfully captures the general shape of this diurnal cycle, but discrepancies remain in the amplitude and timing. The model slightly underestimates daytime mole fractions, with a minimum value that is 1.22 ppm lower and occurs one hour earlier than the observations (at 15:00 LT). The peak $CO_2$ mole fractions in WRF-GHG are also reached at 7:00 LT, but overestimated by 3.36 ppm. Furthermore, this peak is less distinct as in the observations, where the model remains relatively stable between 3:00 and 8:00 LT.

L276-280: While entirely true and important for $CO_2$ – what is the relevance of this part here? Feels disconnected, while it should be part of broader PBL-biosphere dynamics discussion. Also - it's more discussion than results.

We have removed this part from the section and included it into a broader discussion on PBL dynamics further in the manuscript (in the new Discussion subsection on remaining uncertainties:).

Several factors that were not explicitly tested in this study may still contribute to the remaining biases in simulated near-surface $CO_2$. One potential source of uncertainty in many regional modeling studies is the coupling between planetary boundary layer (PBL) dynamics and biogenic $CO_2$ fluxes, a relationship often referred to as the atmospheric $CO_2$ *rectifier effect* (Larson & Volkmer, 2008). Since both processes are driven by solar radiation, they interact nonlinearly throughout the diurnal cycle: daytime $CO_2$ minima result from enhanced turbulent mixing and photosynthetic uptake, whereas nighttime maxima arise from stable stratification and ecosystem respiration. Consequently, accurate simulations require both realistic net ecosystem exchange (NEE) and reliable PBL dynamics. However, in our case, the companion study on $CH_4$ did not reveal any systematic biases in the mean diurnal cycle, suggesting that PBL processes are reasonably well represented here and unlikely to be the dominant cause of the remaining $CO_2$ discrepancies. Nevertheless, it is well known that the choice of PBL schemes can have a substantial influence on simulated tracer concentrations (Díaz-Isaac et al., 2018; Feng et al., 2019; Kretschmer et al., 2014; Yu et al., 2022), and uncertainties are generally amplified under under weak mixing conditions, such as during the night (Maier et al., 2022). Minor deviations in modeled turbulence or nocturnal stability could therefore still contribute to the observed nighttime biases.

L284: "Applying these values to China likely introduces regional mismatches." - Please discuss why. This is part of rationale for this sens. test.

While VPRM parameters are defined per vegetation class (or plant functional type (PFT)), transferring fitted parameters across regions can introduce significant mismatches in NEE estimates. This is because ecosystems classified under the same PFT often differ in dominant species, physiological traits, and phenological patterns. For example, evergreen needleleaf forests in the U.S. and China may host different conifer species with distinct photosynthetic and respiration responses. Additionally, regional climate conditions such as temperature, radiation, and soil moisture affect light-use efficiency and respiration sensitivity. Land use history, forest age, and disturbance regimes further contribute to variability in carbon fluxes. VPRM parameters are typically calibrated using local flux tower data, and studies have shown that regional optimization improves model performance (Mahadevan et al., 2008; Seo et al., 2024). Using parameters from another region assumes the ecosystems and climate are similar, which often isn't true. We will add the following explanation to the manuscript:

Applying these values to China likely introduces regional mismatches, as differences in dominant species, climate conditions, and land use history can significantly alter ecosystem carbon dynamics even within the same vegetation class (Mahadevan et al., 2008; Seo et al., 2024).

L288: "In WRF-GHG..." – Lack of $CO_2$ biogenic flux in those cells possibly has a major impact on low-level mole fractions in the vicinity of Beijing, strongly affecting model performance. How large are the areas

within reasonable distance (100km) around the station that provide NEE fluxes of zero (classified in VPRM as urban). Land-use data from SYNMAP and Copernicus LCC are available to the authors. It could give some insight into the magnitude of the issue.

NEE in WRF-GHG is calculated as a weighted average across vegetation classes, with each class assigned a fractional coverage based on the underlying land cover dataset. Figure A3 in the preprint displays the dominant vegetation class per grid cell in domain d03. However, there are only a few cells categorized as 100% no vegetation, resulting in a zero flux. To illustrate this, we created a map with the fraction values of this class for both land cover datasets, see Fig L. Within a 100 km radius of the site, the SYNMAP dataset identifies 76 grid cells (out of 3522) with 100% no vegetation, while the Copernicus dataset shows none, although 10 cells exceed a 99% fraction. While SYNMAP may lead to an underestimation of NEE in urban areas due to its coarser classification, the Copernicus dataset offers a more detailed land cover representation, which we believe is sufficient to capture NEE signals from urban regions.

L298: "Here, the focus is instead..." - With the rectifier effect important for $CO_2$, wouldn't it be wise to study PBL dynamics in detail? This is important – because of this connection we cannot be sure about the fluxes unless we are sure of PBL. This requires – at the very least - discussion about the accuracy of the PBL in the current scheme, and potential errors that can stem from using the selected PBL scheme. Some conclusions can perhaps be drawn based on available literature without the need to run extra experiments, but it is opinion of this reviewer that this topic has been treated too lightly here.

We fully agree that PBL dynamics are critical for interpreting near-surface $CO_2$ mole fractions, and that uncertainties in PBL representation can influence conclusions about surface fluxes. The sentences referenced by the reviewer were intended to clarify why our focus in this manuscript is not on PBL performance: different PBL schemes available in WRF-GHG were already tested and discussed in Part 1 of this two-part study (Callewaert et al., 2025). In that work, we also demonstrated that key aspects of the diurnal cycle of near-surface $CH_4$ were reasonably well captured by the model. The nighttime overestimation of near-surface $CO_2$ found in the present manuscript is not observed for $CH_4$ at the same site, supporting the interpretation that this discrepancy more likely reflects limitations in $CO_2$ fluxes rather than inaccuracies in PBL dynamics. As a result of restructuring the manuscript and other reviewer's comments, we expanded the Discussion section with a dedicated subsection on remaining uncertainties, with particular emphasis on the influence of PBL dynamics on the $CO_2$ simulations at Xianghe (see above). We believe that these additions clarify the limits of the current analysis while keeping the scope of the paper appropriately focused.

L301: See specific major comment 2.
We have renamed the sensitivity experiments as suggested.

L349: "In contrast, ..." – Wording. This sentence also shows that the respiration is higher, so there is no contrast (?).
Indeed, we have omitted these words.

L387: "specific adjustment – such as (...) – can substantially improve model performance" – I'm not convinced that the change is substantial. In MBE for LC and PARAM yes, but other metrics don't show that much of a change. And Table 6 the results show that MBE in most cases that actually deteriorates (!) when emission profiles are applied. It's more grounded in data to say that "the model becomes more realistic in its setting", but substantial improvement is only seen when PARAM simulation is compared to BASE, in the afternoon data (easiest in PBL). I do not really trust nighttime comparisons due to improper selection of hours for analysis – see comment above.

Indeed, it is not that evident that applying emission profiles improves model performance, as in some periods the LC results are worse than PROF. As suggested in a comment above, we have modified the definition of the nighttime and afternoon periods, and the new table is given in 4. Moreover, we have moved the discussion of the sensitivity experiment results and limitation to a new subsection, whereby this referred sentence was removed from the manuscript. For completion, we hereby provide the content of this new section:
**Emission height**
In Sect. 3.2, we showed that applying vertical profiles to anthropogenic emissions improved the near-surface $CO_2$ simulations at Xianghe, substantially reducing the observed nighttime overestimation in the BASE experiment. These findings are consistent with Brunner et al., 2019 and Peng et al., 2023, who highlighted the critical role of emission height in determining near-surface $CO_2$ mole fractions, particularly under weak mixing conditions. The strong impact on nighttime simulations at Xianghe is likely driven by the proximity of strong point sources (Fig. 6a,b), where the effective release height determines whether emissions remain trapped below or mix above the shallow nocturnal boundary layer.

Nevertheless, some discrepancies with observations remain. In May, for instance, BASE agrees better with observations than PROF, despite May being selected due to poor agreement in the one-year BASE simulation. This apparent contradiction reflects both the short (two-week) duration of the sensitivity runs and the temporal variability of model performance: the main mismatch in BASE occurred in early May, which was not included in the experiments. A full-year sensitivity study would better capture meteorological variability and provide a more robust evaluation, but was not feasible here. The poorer performance of PROF in the second half of May, and the larger MBE in July, remain unexplained, and suggest that further assessment is needed to determine whether the vertical profiles from Brunner et al., 2019 are appropriate for China and consistent with the sectoral and spatial patterns of the emission inventory used in this study.

**Land cover representation**

Replacing the land-cover dataset with the Copernicus product systematically reduced simulated NEE and nighttime $CO_2$ mole fractions at Xianghe. This weakening of the biospheric signal is consistent with the larger fraction of non-vegetated land in the Copernicus compared to SYNMAP, leading to smaller VPRM-driven fluxes. The differences are negligible in December and March, when biospheric activity is minimal. While implementing the Copernicus map does not uniformly improve agreement with observations and slightly worsens performance in some periods, it provides a more realistic land-cover representation and is therefore recommended for future regional experiments.

**VPRM parameterization**

Applying the VPRM parameters from Glauch et al., 2025 produces higher NEE, primarily due to larger $\alpha$ and $\beta$ values that enhance respiration rates (Table A1). Gross ecosystem exchange (GEE) remains similar between the two configurations, indicating that the net effect is primarily driven by increased respiration rather than changes in photosynthetic uptake. Across all experiments and periods, the PARAM configuration shows the closest agreement with observations, though residual discrepancies suggest that additional model errors remain. A key limitation is the absence of a standardized VPRM parameter set optimized for China. The only known regional calibration, by Dayalu et al., 2018, introduces seasonal crop subtypes but requires detailed, time-varying land-cover input that is not readily available. Moreover, Seo et al., 2024 reported that the parameter values adopted here (Li et al., 2020) outperform those of Dayalu et al., 2018 over East Asia, supporting our choice. Recent studies also show that including soil moisture effects in VPRM can improve simulated NEE (Gourdji et al., 2022; Segura-Barrero et al., 2025). However, implementing such modifications would require additional region-specific parameters and could introduce further uncertainty. While the Glauch et al., 2025 parameter set reduces model–observation errors, future work should focus on dedicated VPRM calibration using multi-site eddy-covariance data to develop parameter sets representative of Chinese ecosystems.

L418: "thereby reduced" - Wording: the importance was not "reduced", since it was always smaller for $XCO_2$. Please rephrase.
Indeed, we have rephrased this to: In contrast to $XCO_2$, near-surface $CO_2$ was more strongly influenced by local sources, with a mean tracer enhancement of 26.78 ppm, resulting in a smaller relative importance of boundary condition errors.

L421-422: "Likely causes include..." – what about PBL bias in the model?
As noted earlier, we now include an expanded discussion of uncertainties related to PBL dynamics in the revised Discussion section. In the present work, however, our sensitivity experiments focus specifically on uncertainties in $CO_2$ fluxes. The sentence in the Conclusions therefore highlights the factors directly assessed in our analysis. By using "include" we acknowledge that additional contributors, such as PBL biases, exist and are discussed in the relevant subsection. Since PBL performance is not a central focus of this study, we prefer not to add it explicitly to the Conclusions.

L424-426: "Adjustments to the VPRM vegetation parameters substantially affected near-surface mole fractions, underscoring the critical role of appropriate parameter selection—especially in the absence of a standardized VPRM configuration for China." – please provide numbers here.

We included ... , with differences up to 10 ppm, ... in that sentence, highlighting the difference found in the biogenic $CO_2$ tracer in July between the LC and PARAM experiments.

L430: "while emissions dominate" - anthropogenic or biospheric+anthropogenic? Please be precise.

As this specific sentence summarizes the monthly mean contribution of the biogenic tracer to near-surface $CO_2$ at Xianghe, we refer to biogenic emissions. However, to avoid confusion, we suggest to replace this with 'respiration': At the surface, biospheric uptake is only seen in August (–6.76 ppm), while respiration dominates the rest of the year (+2.69 ppm on average).

**References**

[revised manuscript text omitted]

Figure G: Updated Fig. A1 with a new column showing NEE (mol km$^{-2}$ h$^{-1}$), and with 800 hPa geopotential height contour lines added to the first two columns.

[Figure]

Figure H: Extended period of Fig. A1 with a new column showing NEE (mol km$^{-2}$ h$^{-1}$).

[Figure]

Figure I: Mean diurnal cycle of simulated (a) 2m temperature, (b) 10m wind direction, (c) 10m wind speed, (d) planetary boundary layer height and (c) in situ $CO_2$ mole fractions at Xianghe with respect to local time. The dotted line indicates 00:00 UTC, while the shaded areas represent the interquartile range.

[Figure]

Figure J: Updated Figure 2 from the preprint, showing the time series of the observed (black) and bias-corrected simulated (red) (a) $XCO_2$ and (b) insitu $CO_2$ mole fractions, and there differences in (c) and (d), at the Xianghe site. All hourly simulated data points are shown.

[Figure]

Figure K: Figure showing the original (not bias corrected) XCO$_2$ model values, with all hourly simulated data points. This figure will be palced in the appendix.

[Figure]

Figure L: Relative fraction of 'No vegetation' VPRM class in WRF-GHG domain d03 (3 km) for (a) 1-km SYNMAP and (b) 100-m Copernicus Dynamic Land Cover Collection 3 (epoch 2019). The black cross indicates the location of the Xianghe site, while the circle represents a radius of 100 km.